# Crop–Weed Introgression Plays Critical Roles in Genetic Differentiation and Diversity of Weedy Rice: A Case Study of Human-Influenced Weed Evolution

**DOI:** 10.3390/biology12050744

**Published:** 2023-05-19

**Authors:** Xing-Xing Cai, Zhi Wang, Ye Yuan, Li-Hao Pang, Ying Wang, Bao-Rong Lu

**Affiliations:** Ministry of Education Key Laboratory for Biodiversity Science and Ecological Engineering, School of Life Sciences, Fudan University, Shanghai 200438, China; xxcai@fudan.edu.cn (X.-X.C.); 18110700017@fudan.edu.cn (Z.W.); 20110700002@fudan.edu.cn (L.-H.P.); wang_y@fudan.edu.cn (Y.W.)

**Keywords:** agroecosystem, genetic diversification, *indica*–*japonica* rice, molecular fingerprint, natural hybridization, *Oryza sativa*

## Abstract

**Simple Summary:**

To generate knowledge on how human activities influence plant evolution in agroecosystems, we analyzed allelic introgression from *japonica* rice varieties into the *indica* type of weedy rice, and the impact of crop-to-weed introgression on the genetic differentiation and diversity of the weedy populations in Jiangsu Province of China, based on InDel (insertion/deletion) and SSR (simple sequence repeat) molecular fingerprints. Results from these analyses indicated a positive correlation between increased introgression from *japonica* rice varieties and genetic differentiation in weedy rice. In addition, increased crop-to-weed introgression formed a parabola pattern of dynamic genetic diversity in weedy rice. Our case study indicated that human activities such as the frequent change in crop varieties can influence the evolution of their conspecific weeds through crop-to-weed introgression, which promotes their genetic differentiation and dynamics of genetic diversity in agroecosystems.

**Abstract:**

As an important driving force, introgression plays an essential role in shaping the evolution of plant species. However, knowledge concerning how introgression affects plant evolution in agroecosystems with strong human influences is still limited. To generate such knowledge, we used InDel (insertion/deletion) molecular fingerprints to determine the level of introgression from *japonica* rice cultivars into the *indica* type of weedy rice. We also analyzed the impact of crop-to-weed introgression on the genetic differentiation and diversity of weedy rice, using InDel (insertion/deletion) and SSR (simple sequence repeat) molecular fingerprints. Results based on the STRUCTURE analysis indicated an evident admixture of some weedy rice samples with *indica* and *japonica* components, suggesting different levels of introgression from *japonica* rice cultivars to the *indica* type of weedy rice. The principal coordinate analyses indicated *indica*–*japonica* genetic differentiation among weedy rice samples, which was positively correlated with the introgression of *japonica*-specific alleles from the rice cultivars. In addition, increased crop-to-weed introgression formed a parabola pattern of dynamic genetic diversity in weedy rice. Our findings based on this case study provide evidence that human activities, such as the frequent change in crop varieties, can strongly influence weed evolution by altering genetic differentiation and genetic diversity through crop–weed introgression in agroecosystems.

## 1. Introduction

Evolution is one of the most important concepts in biology, in which four essential forces, including mutation or genetic variation, selection, gene flow or introgression, and genetic drift, drive the evolutionary process [1,2,3,4,5,6,7,8,9,10]. As an important driving force, introgression plays an essential role in shaping the evolution of plant species [5,6,7,8,9,10,11,12,13]. Novel alleles can be transferred from one plant species/population into another genetically diverged or distinct species/population through introgression [5,6,7,8,10]. This process can cause considerable changes in allele frequencies and consequently influence the evolution of the recipient species/populations [5,6,8,13]. When the cross-compatible plant species/populations come into contact, bidirectional or unidirectional introgression is likely to occur naturally through pollen-mediated gene flow [5,8]. It is estimated that at least 25% of plant species are involved in introgression from the same or different species [9]. Studies have also indicated that introgression is more effective in bringing genetic variation into a recipient plant population than mutations [5,6,10]. In addition, introgression can alter the genetic diversity of a recipient population in two ways: increases in genetic diversity by the transfer of novel alleles from other species/populations [5,6,7,10,12], or in contrast, losses of genetic diversity by genetic swamping [6,13].

Domesticated plant species, which are commonly referred to as crops, and their phylogenetically close wild relatives provide excellent cases for studying the influences of introgression on evolution, because of their intimate relationships, allowing natural introgression to occur among crops and their wild relatives [7,8,12,14,15,16,17]. Hübner et al. sequenced the total genomes of cultivated and wild sunflowers to determine the origin of genetic variation in cultivated sunflowers through wild-to-crop introgression, where ~1.5% genetic variation in cultivated sunflowers had its origin from introgression [14]. In addition, the estimate of introgression and its evolutionary impact involving crop species and their conspecific weed is very practical because of the following advantages: firstly, as the same biological species, natural introgression between a crop species and its conspecific weed occurs frequently without reproductive barriers [8,12,14,15,16,17,18]; secondly, unidirectional introgression from a crop to its conspecific weed occurring in the same agricultural field can be easily detected through the identification of proper molecular fingerprints [8,15,19,20]; and thirdly, the cultivation histories and farming styles of a crop species are readily available [17,18]. Altogether, these advantages provide an ideal system to trace the evolutionary process of weedy species under the influence of human activities.

In fact, a great number of studies have confirmed frequent and extensive introgression of crop-specific alleles into weedy populations in agroecosystems, including weedy sorghum (*Sorghum halepense* L.), weedy sunflowers (*Helianthus annus*), and weedy rice (*Oryza sativa* f. *spontanea*) [8,15,16,17,20]. Such introgression had considerable influences on genetic diversity and adaptation of the recipient weedy populations, which created more weed problems, and also on the extinction of small populations of wild relatives [7,13,16,20]. Therefore, it is possible to reveal the consequences and underlying mechanisms of crop-to-weed introgression and its evolutionary impact by studying a conspecific weed and its cooccurring crop species. Crop-specific alleles can accumulate in weedy populations by introgression from different crop varieties through time, which may result in genetic differentiation and increased adaptability of the weedy populations [18,19,20,21,22,23,24]. However, the knowledge on how crop-to-weed introgression influences genetic differentiation and diversity of the receipt weedy populations is still limited. 

Weedy rice is a typical conspecific weed in the genus *Oryza* (*Poaceae*), infesting worldwide cultivated rice (*Oryza sativa* L.) fields and causing considerable rice yield losses around the world [25,26,27]. Similar to its cultivated counterpart, weedy rice is also differentiated into *indica* and *japonica* types, depending on its cooccurring rice varieties at the large scales of its geographical distribution [21,23,25,27]. Generally, the *indica* type of weedy rice infests the tropical or subtropical rice cultivation regions, such as in southern China, South, and Southeastern Asian countries [27,28,29,30], whereas the *japonica* type of weedy rice occurs in the temperate rice cultivation regions, such as in northern and northeastern China, the United States, and southern Europe [21,27,29,30,31,32]. Weedy rice populations with the origin from de-domestication are genetically similar to their cooccurring cultivated rice varieties, showing *indica–japonica* genetic differentiation [22,32]. Usually, long-term cooccurring of weedy rice populations with rice varieties has resulted in their similar morphological and physiological characteristics, most likely through natural gene flow or introgression [27,28]. Therefore, weedy rice provides an ideal system for studying the evolutionary impact of crop-to-weed introgression, because human activities such as the change in rice varieties and cultivation styles happen rapidly in a particular rice planting region [17,18,19,20,21,22]. 

Jiangsu Province (JS) is an important rice planting and production region in China, where *indica* rice varieties were prominently cultivated traditionally [33,34,35,36,37]. The gradual replacement of rice seedling transplanting by direct seeding in JS for the past decades largely promoted the emergence and infestation of weedy rice in rice fields, even though farmers have used commercial and certified rice seeds, and applied herbicides to control weeds [29,33,34]. Historically, no natural distribution of wild *Oryza* species has been reported in this province. Studies have indicated that JS weedy rice has its de-domestication origins, meaning that JS weedy rice has evolved from the domesticated rice varieties [38,39,40,41], such as the origins of weedy rice from other rice planting regions (e.g., northeastern China) where no wild *Oryza* species are distributed either [21,29,31]. Given that no wild rice species are naturally distributed in these regions, weedy rice there is genetically similar to the cultivated rice, rather than wild *Oryza* species (e.g., *O. rufipogon* and *O. nivara*) [34,38,39,40,41,42]. Similar to the characteristics of their co-occurring rice varieties, weedy rice populations found in JS were essentially the *indica* types [23,34]. Since the end of the 1950s, *japonica* rice varieties were gradually cultivated in this province due to the favorable taste and high yielding of *japonica* rice varieties [35,36]. Now, *japonica* rice varieties are prominently cultivated in JS [33,37]. As a consequence, some weedy rice individuals with *japonica* characteristics are identified in the rice planting regions of this province [23,34,42]. Obviously, the introgression of *japonica*-specific alleles from *japonica* rice varieties into weedy rice with the *indica* genetic background has played an important role in changing *indica–japonica* characteristics of weedy rice in JS where the rapid change from planting *indica* to *japonica* rice varieties has taken place. Apart from observing the *japonica* type of weedy rice in JS, our knowledge concerning *japonica*-crop-to-*indica*-weed introgression in the evolution of weedy rice is still limited. Therefore, investigating the cultivation history of *indica* to *japonica* rice varieties associated with introgression, genetic differentiation, and diversity of weedy rice may provide a deep insight into the role of crop-to-weed introgression in plant evolution. 

In this case study, we used the InDel (insertion/deletion) molecular markers [43,44] to determine the genetic differentiation of weedy rice associated with the extent of crop-to-weed introgression in JS. We also applied microsatellite (simple sequence repeats, SSR) molecular markers to examine the genetic diversity pattern of weedy rice associated with the extent of crop-to-weed introgression. The primary objectives of the study were to (1) determine the historical changes in rice cultivation associated with *indica* to *japonica* varieties in JS; (2) examine whether introgression of *japonica* alleles from cultivated rice affects the genetic structure of weedy rice; (3) and assess the influences of crop-to-weed introgression on genetic differentiation of weedy rice and on the patterns of genetic diversity in weedy rice. The generated knowledge will facilitate our understanding of the impact of human-influenced evolution on weedy species in agroecosystems.

## 2. Materials and Methods

### 2.1. Historical Changes in Rice Cultivation from Indica to Japonica Varieties in Jiangsu Province

The change in planting areas from *indica* to *japonica* rice varieties in Jiangsu Province (JS) was estimated mainly based on the published statistical data, including Jiangsu Agricultural Statistics (1949–1987), Economic Data of Rural Jiangsu Province (1988–1998), and Statistical Yearbook of Rural Jiangsu Province (2000–2017) [45,46,47]. In addition, the published books, including The Science of Rice Cultivation in Jiangsu Province, The History of Agricultural Development in Jiangsu Province, and The Documents of Jiangsu Provincial Annals (Agricultural) [48,49,50], were also used to estimate the historical change in rice varieties cultivated in JS. To demonstrate the pattern of the spatial-temporal changes in *japonica* rice cultivation in JS, we calculated the average areas (by 1000 hectares, kha) of *japonica rice* cultivation for every 10 years, based on each county as a unit from the 1950s to 2020s. The obtained average data of *japonica* rice cultivation areas were visually presented in GIS maps with the different units in JS, using the software ArcGIS ver.10.2 [51]. 

### 2.2. Collection of Plant Materials

Mature seed samples were collected from a total of 36 natural weedy rice populations during October 2020~2021 across the rice planting regions in JS (Figure 1 and Appendix A). The density of weedy rice occurring in the JS rice fields was about 0.5–1.0 plant per 100 m^2^. For sample collection, we included 30~32 randomly selected individuals (samples) from each weedy rice population with the spatial distances >10 m in a sampling rice field (about 6000 m^2^). In comparison, mature seed samples of the accompanying rice varieties in the same fields were also collected. The spatial distances between the collecting sites (fields) for each weedy rice population were >10 km. In addition, mature seed samples from three weedy rice populations each in Guangdong Province (GD) and northeast China (NEC) were collected to represent the *indica* and *japonica* types of weedy rice, respectively (Appendix A). Furthermore, mature seeds of 13 typical *indica* and 13 typical *japonica* rice varieties from various sources identified by “InDel molecular index” [43] were also included as the references for further analyses (Appendix A).

### 2.3. DNA Extraction, Amplification, and Genotyping

Seeds of weedy rice and cultivated rice were germinated in an illuminated incubator (Percival Scientific, Perry, IA, USA) (25 ± 3 °C) with alternating light/dark (16/8 h). The total genomic DNA was extracted from the 14-day-old fresh seedlings following a modified CTAB protocol [52]. 

Thirty-eight *indica–japonica* specific InDel primer pairs (Appendix A) [43,44], distributed on both arms of each of the 12 rice chromosomes, were selected to determine *indica* and *japonica* characteristics of all the weedy and cultivated rice samples. In addition, 47 SSR primer pairs (Appendix A) from the rice genome distributed across the 12 chromosomes were selected from the Gramene Markers Database (https://archive.gramene.org/markers/, accessed on 20 April 2023) [53] to analyze genetic diversity of weedy rice populations. All the forward primers of the selected InDel and SSR primer pairs were labeled with one of the following fluorescent dyes: FAM (blue), HEX (green), ROX (red), and TAMRA (black), respectively. 

Polymerase chain reactions (PCR) were performed in a total volume of 20 µL containing 1 × PCR buffer (with Mg^2+^), 200 µM dNTPs, 4 µM of each primer, 0.5 U Taq polymerase, 40 ng template DNA, and ddH_2_O to the final volume. Reaction conditions comprised an initial denaturation step for 4 min at 94 °C, followed by 30 cycles of 30 s at 94 °C, 30 s at 52~58 °C, and 30 s at 72 °C, and a final extension step at 72 °C for 10 min. According to the size of the PCR products, labeled PCR products of 3~5 InDel or SSR primer pairs were mixed together in a ratio of FAM:HEX:ROX:TAMRA = 1:3:1:3, then were electrophoresis separated on ABI 3730xl Analyzer (Applied Biosystems, Waltham, MA, USA). For genotyping, the separated InDel or SSR fragments of each sample were scored using the software GeneMapper version 4.1 (Applied Biosystems, Waltham, MA, USA).

### 2.4. Data Analysis

#### 2.4.1. *Indica–Japonica* Characterization of Weedy Rice and Cultivated Rice

The frequency of *japonica*-specific alleles (*F_j_*) of each weedy and cultivated rice sample was calculated based on the “InDel molecular index” [43]. For each InDel locus, the typical *indica* rice variety (93–11) and the *japonica* rice variety (Nipponbare) were used as the reference to determine the homozygote *indica* (II), *japonica* (JJ), and heterozygote (IJ) genotypes, respectively (for details, see Lu et al. [43]). In this study, the *indica–japonica* characterization of all weedy and cultivated rice samples was determined as the *indica* type (*F_j_* < 0.25), intermediate type (0.25 ≤ *F_j_* < 0.75), and *japonica* type (*F_j_* ≥ 0.75), respectively, based on the average values of *F_j_* or “InDel molecular index” of the 38 InDel loci [43,44]. 

#### 2.4.2. Estimate of Crop-to-Weed Introgression Using the Frequency of *Japonica*-Specific Alleles (*F_j_*)

Introgression of the *japonica*-specific alleles was examined based on the InDel data matrix, including JS weedy rice that originally had an *indica* genetic background and the reference rice samples (26 typical *indica* and *japonica* verities), using the software STRUCTURE v.2.3.4 [54]. The STRUCTURE analysis was conducted with the admixture model and the correlated allele frequency model among groups, with an initial burn-in run of 100,000 steps followed by 200,000 MCMC iterations. The group number (K) was set from 2 to 8 with 10 runs for each K value. The appropriate K value was determined by calculating the value of ΔK described in Evanno et al. [55] using the Structure Harvester (https://taylor0.biology.ucla.edu/structureHarvester/ accessed on 20 April 2023). 

To estimate the possible application of the frequency of *japonica*-specific alleles (*F_j_*) for measuring the levels of crop-to-weed introgression from *japonica* rice varieties, we analyzed the correlation between the *F_j_* values [43] and the proportion of introgressed *japonica* genetic components (alleles) derived from the “inferred ancestry of individuals” cluster data matrix generated in the STRUCTURE analysis [54]. Results of the correlation between the *F_j_* values and the proportion of introgressed *japonica* genetic components (alleles) were visualized by a linear model of Pearson’s correlation method using Prism v.8.0.2 [56]. 

#### 2.4.3. Correlation between Genetic Differentiation and Crop-to-Weed Introgression in JS Weedy Rice 

To determine the *indica–japonica* genetic differentiation pattern of JS weedy rice, we performed a principal coordinate analysis (PCoA) of the genotypic data matrix based on the 38 InDel molecular fingerprints, using the software GenAlEx v.6.5 [57]. The data matrix included weedy rice samples from JS, GD, and NEC, in addition to the typical *indica* and *japonica* rice varieties as references. Results from the first two principal coordinates of all studied samples were graphed in a 2-dimensional scatterplot to illustrate the genetic differentiation of JS weedy rice samples. 

To estimate correlations between genetic differentiation and crop-to-weed allelic introgression of weedy rice, we established 30 ideally sampled weedy rice populations (ISWPs), each containing 16 samples randomly drawn from the JS weedy rice pool with a total of 1116 samples. These populations represented the low (*F_j_* < 0.25), middle (0.25 ≤ *F_j_* < 0.75), and high (*F_j_* ≥ 0.75) levels of introgression, respectively. Consequently, there were 10 low-introgression, 10 middle-introgression, and 10 high-introgression ISWPs established for further correlation analyses. In addition, we randomly selected eight naturally sampled weedy rice populations (NSWPs) from the field sampled JS weedy rice populations based on their *F_j_* values, ranging from 0.05 (lowest) to 0.34 (highest), to analyze the relationships between genetic differentiation of weedy rice and crop-to-weed allelic introgression revealed by ISWPs.

The pairwise Wright’s *F_st_* values [58] were calculated to represent the genetic differentiation (*F_st_*) between ISWPs based on the genotyping data matrixes of both InDel and SSR molecular fingerprints, respectively, using the software GenAlEx v.6.5 [57]. Moreover, the pairwise differences in the *F_j_* values were calculated to represent the differences in introgression (*F_j-d_*) between ISWPs based on the same data matrixes. The correlation between genetic differentiation and crop-to-weed allelic introgression was calculated based on the obtained *F_st_* and *F_j-d_* values of the 30 ISWPs and eight NSWPs, respectively. The obtained correlation was visualized using Pearson’s correlation method (linear model) packaged in the software Prism v.8.0.2 [56]. 

#### 2.4.4. Correlation between Genetic Diversity and Crop-to-Weed Introgression in JS Weedy Rice 

To estimate the correlation of genetic diversity (represented by Nei’s expected heterozygosity, *H_e_* and Shannon’s information index, *I*) [59,60] with the level of crop-to-weed introgression (*F_j_*), we calculated the two genetic diversity parameters using the same ISWPs and NSWPs (see the Section 2.4.3), respectively, based on the genotyping data matrices of both InDel and SSR molecular fingerprints. The association of the genetic diversity parameters (*H_e_* and *I*) with *F_j_* values was obtained using the quadratic equation fitting regression analysis for the ISWPs. The obtained results were visualized using a nonlinear model packaged in the software Prism v.8.0.2 [56]. In addition, we also analyzed the association of the genetic diversity parameters (*H_e_* and *I*) with the *F_j_* values in the eight NSWPs using Pearson’s correlation method (linear model) packaged in the software Prism v.8.0.2 [56] to confirm the correlation pattern obtained based on ISWPs. 

## 3. Results

### 3.1. Rapid Alteration of Rice Cultivation from Indica to Japonica Varieties in Jiangsu Province

The historical archives indicated that the tradition of rice cultivation in Jiangsu Province (JS) was essentially *indica* varieties until the early 1950s, when *japonica* rice varieties were introduced to this province due to their favorable commercial quality (tastes) and high yield. Statistical data between the 1950s and 2010s evidently demonstrated a relatively rapid change in rice cultivation from *indica* to *japonica* rice varieties in this province (Appendix A and Figure 2). Starting from the end of the 1950s, the cultivation areas for *japonica* rice varieties gradually expanded into the areas where the *indica* varieties were originally cultivated in this province within only a few decades. Consequently, *indica* rice varieties were almost completely replaced by *japonica* rice varieties by the end of the 1990s, particularly in some rice cultivation areas, such as the northeast, east, and south part of this province (Figure 2).

The statistical data also indicated that the alteration of cultivation from *indica* to *japonica* rice varieties started in the southern part of JS. Then, the cultivation areas of *japonica* rice varieties were gradually extended northward to cover the rice cultivation areas in the whole province. The cultivation areas for *japonica* rice varieties increased from only 3.4 kha (1000 hectares) in the 1950s to 2015.3 kha in the 2010s. In contrast, the cultivation areas for *indica* rice varieties dramatically decreased from 1988.5 kha in the 1950s to 290.0 kha in the 2010s (Appendix A). Therefore, the different durations of rice cultivation for *japonica* varieties in different areas could cause the different levels of *japonica*-specific allelic introgression into the *indica* type of weedy rice cooccurring with the *japonica* rice varieties in these areas.

### 3.2. Patterns of Introgression from Japonica Rice Varieties to Weedy Rice

In general, results based on the calculated “InDel molecular index” clearly indicated that each of the examined cultivated rice and weedy rice samples from Guangdong Province (GD), northeast China (NEC), and JS had their distinct *indica*, *japonica*, or *indica–japonica* (determined as intermediate) characteristics (Appendix A). This characterization was based on the “InDel molecular index” or examined the frequency of *japonica*-specific alleles (*F_j_*) at the 38 InDel (insertion/deletion) loci across the rice genome. The *F_j_* values of the typical *indica* rice varieties and weedy rice samples from GD were equal to 0 or close to zero, whereas the *F_j_* values of the typical *japonica* rice varieties and weedy rice samples from NEC were equal to one or close to one (Appendix A). These results suggest that the reference cultivated rice and weedy rice samples from GD or NEC rice cultivation regions had identical and unique *indica* or *japonica* characteristics, respectively, which set up an ideal standard for a comparison of the *indica–japonica* characteristics of JS weedy rice samples. However, the *F_j_* values of the 1116 weedy rice samples from JS ranged from 0 to 0.97 (Appendix A); although, most of the samples still showed their *indica* characteristics (Appendix A). In the JS weedy rice pool, ~91% were the *indica* type (*F_j_* < 0.25), ~7% were the intermediate type (0.25 ≤ *F_j_* < 0.75), and ~2% were the *japonica* type (*F_j_* ≥ 0.75) (Table 1). In addition, most (~81%) cooccurring rice cultivars from JS belonged to the *japonica* type (Appendix A). These results clearly indicated the introgression of *japonica*-specific alleles from *japonica* rice varieties into the original *indica* type of weedy rice populations in the JS rice cultivation region.

To determine the level of introgression of *japonica*-specific alleles into weedy rice in JS rice fields, where weedy rice was originally composed of the *indica* genetic background, we conducted the STRUCTURE analysis using the admixture model based on the data matrices of the InDel molecular fingerprints. Results from the STRUCTURE analysis demonstrated the distinct genetic components of the typical *indica* (red and blue) and *japonica* (green) rice varieties (references) at the most optimal K value (K = 3) and their neighboring K values (K = 2, K = 4). The findings suggested the distinct genetic components and substantially diverged genetic relationships of the *indica* and *japonica* cultivated rice samples used as references (Figure 3). The distinguishable *indica* and *japonica* genetic components set up an excellent standard for studying the introgression of *japonica*-specific alleles (green) into the *indica* genetic background (red and blue) of weedy rice samples/populations.

As expected, weedy rice samples collected from different locations in JS showed their distinct genetic components (Figure 3). Compared with the genetic components of the references (*indica*: red and blue, *japonica*: green), the weedy rice samples from JS showed both *indica* and *japonica* genetic components (Figure 3). Apparently, most JS weedy rice samples had *indica* genetic components, suggesting their close genetic affinity with their originally cultivated *indica* rice varieties (Figure 3). However, some JS weedy rice samples exhibited an admixture of *indica–japonica* genetic components, indicating the different degree of introgression from *japonica* rice varieties to the *indica* type of weedy rice. In addition, the admixture genetic components of the *japonica* weedy rice also ruled out the possible contamination of *japonica* weedy rice seeds that should not have admixture components. Noticeably, the typical *japonica* rice reference varieties showed a distinctly unique genetic component (green) with nearly no admixture; although, the typical *indica* rice reference varieties showed somehow other genetic components with an extremely low level of admixture at the K = 3 and K = 4 (Figure 3). These results provide opportunities to analyze the level of introgression of the *japonica*-specific alleles in the JS weedy rice samples, most of which had *indica* genetic components.

To determine whether the frequency of the *japonica*-specific alleles (*F_j_*) obtained based on the “InDel molecular index” could be directly used for estimating the level of allelic introgression from *japonica* varieties into the *indica* type of weedy rice, we analyzed the correlation between the obtained *F_j_* values and the ratios of the introgressed *japonica*-component of weedy rice samples extracted from the “ancestry of individuals” cluster data matrix in the STRUCTURE analysis. Results from the correlation analysis indicated a significant positive correlation (*R*^2^ = 0.94, *p* < 0.001) between the *F_j_* values and the ratios of introgressed *japonica* components (Figure 4). This finding suggests that the *F_j_* values calculated based on the “InDel molecular index” (Appendix A) could be used to estimate the level of *japonica*-specific allelic introgression for further analyses, particularly the relationships of crop-to-weed introgression with genetic differentiation and genetic diversity in JS weedy rice based on the *F_j_* values. 

### 3.3. Genetic Differentiation in Weedy Rice Associated with Crop-to-Weed Introgression

To investigate the patterns of genetic differentiation in JS weedy rice samples, we conducted the principal coordinate analysis (PCoA) based on the InDel molecular fingerprints, using typical *indica* and *japonica* rice varieties and weedy rice samples from GD and NEC as references. The PCoA results demonstrated evident genetic differentiation of JS weedy rice samples into *indica* and *japonica* types (Figure 5). Obviously, most weedy rice samples from JS were scattered and were closely associated with the typical *indica* rice varieties and the weedy rice samples from GD, at the negative loads of the first principal coordinate (left in Figure 5). Therefore, these weedy rice samples were likely the *indica* type, which was supported by their low *F_j_* values (<0.25, Table 1). A small proportion of weedy rice samples from JS was scattered among the typical *japonica* rice varieties and the weedy rice samples from NEC (references), at the positive loads of the first principal coordinate (right in Figure 5). Therefore, these weedy rice samples were most likely the *japonica* type, which was also supported by their high *F_j_* values (>0.75, Table 1). Noticeably, some of the JS weedy rice samples were scattered between the typical *indica* and *japonica* types along the first principal coordinate (middle in Figure 5), which were determined as the *indica–japonica* intermediate types with the *F_j_* value between 0.25 and 0.75 (Table 1).

In general, the PCoA results clearly indicate the genetic differentiation of JS weedy rice samples that are scattered between the typical reference *indica* and *japonica* rice varieties. This finding was supported by the gradually increased *japonica*-specific allelic frequency (*F_j_*) in the JS weedy rice samples that should originally be the *indica* type. 

To estimate the correlation between the extent/level of genetic differentiation and crop-to-weed introgression, we calculated the pairwise genetic differentiation (*F_st_*) and differences in allelic frequency (*F_j-d_*), based on the data matrices of InDel (Figure 6a and Figure 7a) and SSR (Figure 6b and Figure 7b) molecular fingerprints. The correlation analysis was conducted based on the 30 ideally sampled weedy rice populations (ISWPs) with their respective low, middle, and high levels of introgression (Figure 6), and eight natural weedy rice populations (NSWPs) (Figure 7) for both InDel and SSR molecular fingerprints. Results based on the Pearson’s correlation analysis showed a significantly positive correlation between genetic differentiation as measured by the pairwise *F_st_* values and crop-to-weed introgression as estimated by the pairwise *F_j-d_* values of ISWPs (*R*^2^ = 0.94–0.96, *p* < 0.001) and NSWPs (*R*^2^ = 0.35–0.73, *p* < 0.01). These results generated from the ISWPs and NSWPs of JS weedy rice suggest that crop allelic introgression would cause considerable genetic differentiation of its conspecific weed. 

### 3.4. Genetic Diversity of Weedy Rice Associated with Crop-to-Weed Introgression

To estimate the correlation/relationship between the level of genetic diversity and crop-to-weed introgression, we calculated Nei’s expected heterozygosity (*H_e_*) and Shannon’s information index (*I*) to represent genetic diversity, based on the data matrices of InDel and SSR molecular fingerprints. In addition, we used the frequency of *japonica*-specific alleles (*F_j_*) to represent the level of crop-to-weed introgression. The correlation analysis was conducted based on the 30 ISWPs with their respective low, middle, and high levels of introgression (Figure 8) for both InDel and SSR molecular fingerprints. Results based on the quadratic equation fitting regression analysis indicated a high degree of fitting (*R*^2^ = 0.98 for *H_e_* and 0.99 for *I*) for the InDel molecular fingerprints (Figure 8a,c). Similarly, the quadratic equation fitting regression analysis indicated a relatively high degree of fitting (*R*^2^ = 0.81 for *H_e_* and 0.78 for *I*) for the SSR molecular fingerprints (Figure 8b,d). 

In addition, the correlation between the level of genetic diversity (*H_e_* and *I*) and crop-to-weed introgression (*F_j_*) was also analyzed based on the eight NSWPs (Figure 9). Results showed that the level of genetic diversity (*H_e_* and *I*) significantly increased with the increases in the *F_j_* values for the InDel molecular fingerprints (Figure 9a,c), when *F_j_* varied between 0.05 and 0.34. However, no significant correlations were observed between the level of genetic diversity (*H_e_* and *I*) and the *F_j_* values for the SSR molecular fingerprints (Figure 9b,d). The relationship between the level of genetic diversity and crop-to-weed introgression (represented by the *F_j_* values) generated based on these results from NSWPs generally agreed with those revealed from ISWPs, particularly for InDel molecular fingerprints.

## 4. Discussion

### 4.1. The Change in Rice Varieties Greatly Influences Indica–Japonica Characteristics of Weedy Rice through Crop-to-Weedy Introgression

Our results in this study based on the historical archives clearly demonstrated the pattern of traditional rice cultivation in Jiangsu Province (JS), where *indica* varieties were grown essentially until the early 1950s. Different *japonica* rice varieties were gradually introduced into this province at the beginning of the 1960s because of their favorable commercial quality (taste) and high yield [35,36,37]. After three decades by the 1990s, the *indica* rice varieties were almost completely replaced by *japonica* rice varieties in different rice cultivation areas in this province. As a consequence, weedy rice with *japonica* characteristics was reported to appear frequently in the rice fields of this province [23,33,42]. Apparently, the changing pattern of rice cultivation from *indica* to *japonica* varieties has considerable influences on the genetic compositions of weedy rice as indicated by its *indica* and *japonica* characteristics. Such influences are hypothetically through gene flow or allelic introgression from *japonica* rice varieties to weedy rice individuals. 

In fact, our results showed that most examined JS weedy rice samples (individuals) were the *indica* type (~91%) and the rest of the weedy rice samples were the intermediate and *japonica* types, based on the InDel molecular index [43] or the average frequency of the *japonica*-specific alleles (*F_j_*). These results were obtained based on a relatively large number of weedy rice samples (1116 individuals) collected from 36 populations across the JS rice cultivation areas. This finding indicates the change in characteristics of JS weedy rice gradually from the original *indica* type to the *japonica* and *indica–japonica* intermediate types, associated with the rapid changes in rice cultivation patterns from *indica* to *japonica* varieties in this province. In other words, the rapid alteration of rice cultivation from *indica* to *japonica* varieties promoted the divergence of JS weedy rice characteristics, essentially through crop-to-weed introgression: although, other factors, such as direct-seeding and mechanic harvesting, cannot be completely excluded. The changes in the *indica–japonica* characteristics of JS weedy rice found in this study are similar to those revealed either by phenotypical characterization [42,61] or molecular fingerprinting [23,34], in which weedy rice was composed of the *indica*, *japonica*, and intermediate types, although predominated by the *indica* type. All results clearly indicate a wide range of variation regarding the *indica–japonica* characteristics of JS weedy rice, owing to the rice cultivation change from *indica* to *japonica* varieties.

In this study, we also found that the reference weedy rice populations from NEC were essentially the *japonica* type that was associated strongly with the typical *japonica* rice varieties, whereas those from GD were essentially the *indica* type that was associated intimately with the typical *indica* rice varieties. Our finding is consistent with the previous reports in which weedy rice is genetically closely associated with its cultivated counterparts co-occurring in the same regions [21,23,27,31]. Obviously, the JS weedy rice samples with different genetic backgrounds identified in this study should be the result of successive gene flow or introgression with different types of *indica* and *japonica* rice varieties cultivated during a historical period of time in this province. Successive crop-to-weed gene flow or introgression, coupled with the independent assortment and genetic recombination in the self-pollination process, promoted the admixture of *indica* and *japonica* genotypes in JS weedy rice. 

Results from the STRUCTURE analysis based on the InDel molecular fingerprints in this study clearly demonstrated the change in genetic compositions in JS weedy rice from the original *indica* type to the current *japonica* types with evident admixture genetic components, as indicated by the different bar-plots with three K values (Figure 3). The presence of the *indica–japonica* admixture types of weedy rice in the JS rice cultivation areas confirmed crop-to-weed allelic introgression, because of the accumulation of *japonica* alleles in the admixture types of JS weedy rice samples. The presence of *indica–japonica* admixture types in the JS weedy rice samples can also exclude the possible contamination of these samples as *japonica* weedy rice seeds. This is because the mixed *japonica* weedy rice samples, if any, in the certified commercial *japonica* cultivar seeds, should be present as the pure *japonica* genotype in the STRUCTURE analysis. Somehow, there is a possibility of the contamination of a few weedy rice seeds with the newly introduced *japonica* rice varieties in the actual rice production. Given that the opportunities for inter-crosses between the distantly scattered weedy rice plants are very low, the gradual accumulation of *japonica* alleles in weedy rice is more likely through crop-to-weed introgression [62,63,64,65,66,67,68] from *japonica* rice varieties, rather than by spontaneous mutations or through seed contamination. In addition, the gradual change in JS weedy rice genetic components from the original *indica* type to the admixture and *japonica* types matches the pattern of JS rice variety change in cultivation for the past seven decades (Figure 2). The relatively low proportion of JS weedy rice samples with *japonica*-specific alleles from their cooccurring cultivars can be explained by the inbreeding feature of weedy rice with a relatively low outcrossing rate (~1%) [62,63,64,65,66,67,68]. Thus, we propose that crop-to-weed introgression has played an important role not only in changing the *indica–japonica* characteristics and genetic components of JS weedy rice, but also in shaping the evolution of JS weedy rice as an important driving force. These observations agree with the previous reports concerning weedy rice genetic diversity and evolution [17,18,19,20,21,22]. 

### 4.2. Crop-to-Weed Introgression Impacts Genetic Differentiation and Genetic Diversity in Weedy Rice through Accumilated Crop-Specific Alleles 

Our results clearly indicate that the level of gene flow or crop-to-weed introgression can be measured by the frequency of the *japonica*-specific alleles (*F_j_*) of the weedy rice samples with considerably high confidence (Figure 4). Results based on the PCoA analysis of the InDel molecular fingerprints also demonstrated substantial genetic differentiation of JS weedy rice, which was somehow associated with crop-to-weed introgression caused by the change in rice cultivation for *japonica* varieties. Given that the *F_j_* value of each weedy rice sample can be accurately calculated based on the InDel molecular fingerprinting [43], we analyzed the impact of crop-to-weed introgression on genetic differentiation and genetic diversity of weedy rice. The analysis of such impact can be realized through calculating the correlation between the *F_j_* values and the level of genetic differentiation (*F_st_*), as well as genetic diversity (*H_e_* and *I*), using the ideally sampled weedy rice populations (ISWPs) established according to their levels (low, middle, and high) of introgression determined by the *F_j_* values. 

Further analyses in this study indicated that the level of *japonica*-specific allelic (crop-to-weed) introgression was highly significantly correlated (*R*^2^ = 0.94, *p* < 0.001) with the level of *F_j_* values of the examined weedy rice samples. Therefore, the *F_j_* values can represent *japonica*-specific allelic introgression and be directly utilized to determine the relationships between the level of crop-to-weed introgression and the genetic differentiation of weedy rice. Results obtained based on ISWPs with low, middle, and high levels of introgression, using both the InDel and SSR molecular fingerprints, clearly demonstrated the correlation patterns, in which crop-to-weed introgression substantially prompted the genetic differentiation of JS weedy rice. Interestingly, similar correlation patterns were obtained based on the randomly selected naturally sampled weedy rice populations (NSWPs) with low to middle levels of introgression in JS, although with a much lower level of correlation for the results obtained using the SSR molecular fingerprints. Therefore, we can conclude based on all results from this study that crop-to-weed introgression can significantly promote the genetic differentiation of weedy rice populations with the original *indica* genotype through accumulating *japonica*-specific alleles from its co-existing *japonica* rice varieties. In other words, the gradual introgression of different crop-specific alleles into weedy rice populations can considerably cause their within- and between-population genetic differentiation.

Our results based on the InDel molecular fingerprints also suggest that the level of *F_j_* values was significantly associated with the two independent genetic diversity parameters (Nei’s expected heterozygosity, *H_e_*, and Shannon’s information index, *I*) in both ISWPs and NSWPs (Figure 8 and Figure 9). Noticeably, the polymorphic SSR molecular markers did not show a significant correlation between genetic diversity and introgression in NSWPs, in addition to a comparably lower level of correlation than the dimorphic (*indica–japonica*) InDel molecular markers. The differences in genetic diversity revealed by SSR and InDel molecular markers can easily be explained by the reasons that the formation of genetic diversity in weedy rice is not only determined by crop-to-weed introgression involving *indica–japonica* alleles or characteristics, but also by other types of alleles that are not associated with the *indica* and *japonica* characteristics. Therefore, we propose that, based on this case study, the *F_j_* values that can represent *japonica*-specific allelic introgression can also be used to determine the relationships between the level of crop-to-weed introgression and the dynamics of genetic diversity in weedy rice populations. This conclusion is supported by previous studies, in which an increased level of crop-to-weed introgression can promote rapid changes in genetic diversity patterns in weedy populations [19,20,23,27]. 

### 4.3. Human Activities Can Accelerate the Evolution of Conspecific Weeds in Agroecosystems 

It is well-known that weedy rice infests worldwide rice fields, causing considerable losses in the grain yield and quality of cultivated rice [25,26,27]. As a conspecific weed of cultivated rice, weedy rice evolved rapidly in rice ecosystems to adapt to the weed control and the environmental changes associated with human activities around the globe [18,25,26,27]. Weedy rice has become a great weed problem for rice cultivation around the world, including in the rice-planting regions in China (e.g., Jiangsu Province), which threatens sustainable rice production [33,42,69]. In some regions, a rapid change in the rice ecosystems has taken place, such as the shift from rice transplanting to direct seeding, the application of farming machinery, and the quick change in rice varieties [35,36,37,69]. These human activities in the rice ecosystems could have imposed a strong impact on the evolutionary processes of weeds, including weedy rice, to adapt to the changing environment. This case study that we completed in Jiangsu Province with documented changes in rice cultivation from *indica* to *japonica* varieties for the past few decades reveals the human-influenced evolution of weedy rice in agroecosystems. 

Our results indicated the presence of the *japonica*-specific alleles in JS weedy rice, most likely through crop-to-weed allelic introgression based on the InDel molecular fingerprints. Such introgression is closely associated with the rapid replacement of *indica* rice varieties by *japonica* rice varieties in JS. This finding evidently indicates that human-influenced cultivation changes in rice varieties (from *indica* to *japonica*) alone have already altered the genetic components of the co-occurring weedy rice populations. Our results from the analysis of InDel and SSR molecular fingerprints based on the ISWPs and NSWPs further demonstrated a positive correlation of crop-to-weed introgression, as measured by the *F_j_* values, with the genetic differentiation of weedy rice. In addition, these results also confirmed the close association between the levels of crop-to-weed introgression and genetic diversity both in ISWPs and NSWPs, using the two sets of molecular fingerprints. 

Altogether, these findings demonstrated that human activities or disturbances can significantly influence the genetic differentiation and genetic diversity of weedy rice—a conspecific weed—only through consecutive allelic introgression from its cooccurring cultivated counterparts that have constantly been improved by humans at different periods in time. This conclusion is supported by the large number of *japonica*-specific alleles detected in JS weedy rice, which is associated with the rapid change in rice cultivation from *indica* to *japonica* varieties. Genetic differentiation and genetic diversity are two important elements that are closely associated with the evolution of plant species. Therefore, we consider that human activities or disturbances can promote the rapid adaptive evolution of conspecific weedy rice in rice ecosystems through the change in these elements, which makes the control and management of weedy rice very difficult. 

Such human-influenced rapid evolution of agricultural weeds as revealed in this case study may also be frequently found in many other weedy plant species in agroecosystems [8,15,16,70,71]. Previous studies reported that crop–weed introgression promoted an increase in genetic diversity in cooccurring weed populations [23,28,71,72], due to the frequent changes in newly developed crop varieties around the world [23,24,73]. Very often, these new varieties contained many new alleles/genes, with even transgenes having great evolutionary potential [17,24,73,74]. The results presented in this study provide a convincing case to explain how a conspecific weed can evolve rapidly by accumulated crop alleles from diverse crop varieties through crop-to-weed introgression to promote its genetic differentiation and diversity. Such an impact on the adaptive evolution of conspecific weeds imposes a great challenge for the control and management of these weeds. 

## 5. Conclusions

In this study, we detected many *japonica*-specific alleles in JS weedy rice that should originally be the *indica* type with the *indica* genetic background, based on the insertion/deletion (InDel) molecular fingerprints. The presence of the *japonica*-specific alleles in the *indica* type of weedy rice is most likely the result of gene flow or introgression of alleles from *japonica* rice varieties; although, in the practical rice production, a very low frequency of *japonica* weedy rice contamination might also happen. Such gradual introgression of *japonica* alleles is closely associated with the change in rice cultivation from *indica* to *japonica* varieties in the past few decades. Our results further indicate that crop-to-weed allelic introgression has considerably changed the genetic components of the cooccurring JS-weedy rice populations. Further analyses based on InDel and SSR molecular fingerprinting indicate a significant positive correlation between the levels of crop-to-weed introgression and genetic differentiation in JS weedy rice. Similarly, increased crop-to-weed introgression promoted a change in genetic diversity in weedy rice with a parabola correlation pattern. Altogether, the above findings demonstrate that human activities, such as the change in cultivated rice varieties, can impose a considerable impact on the evolution of its conspecific weed by promoting genetic differentiation and diversity in rice ecosystems. A similar pattern of crop-to-weed introgression promoting genetic differentiation and genetic diversity is likely found in other crops and their conspecific weeds. Therefore, we conclude that, based on this case study, human activities or disturbances may accelerate the adaptive evolution of conspecific weeds through crop-to-weed introgression in agroecosystems, which may impose a great challenge for the control and management of these weeds. 

## Figures and Tables

**Figure 1 biology-12-00744-f001:**
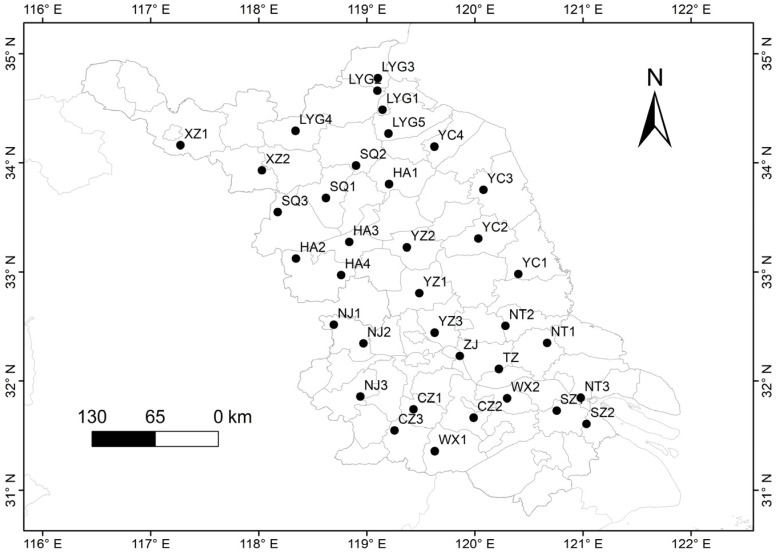
Sampling sites of weedy rice populations (black dots) in Jiangsu Province of China. The identification of the weedy rice populations is provided in Appendix A (Population ID in the 4th column).

**Figure 2 biology-12-00744-f002:**
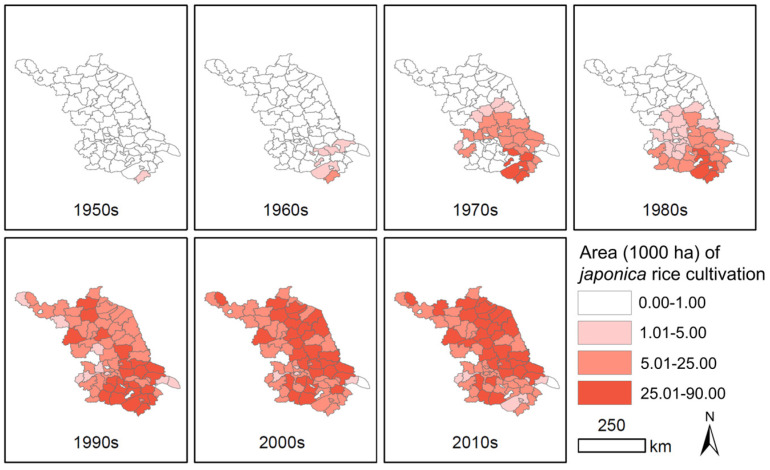
The rapid increases in cultivation areas of *japonica* rice varieties from *indica* rice varieties in Jiangsu Province during the 1950s–2010s. The lines represent the borders of the counties. The color intensity (4 levels) indicates the 10-year average cultivation areas in 1000 hectares (kha) of *japonica* rice varieties in each county: white, 0.00–1.00 kha; light-pink, 1.01–5.00 kha; pink, 5.01–25.00 kha, and red, 25.01–90.00 kha.

**Figure 3 biology-12-00744-f003:**
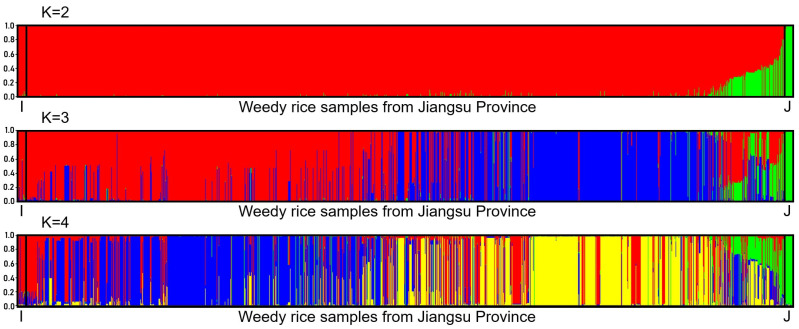
Bar plots indicating genetic components of typical *indica* and *japonica* rice varieties (references) and weedy rice samples from Jiangsu Province based on the STRUCTURE analysis of the InDel molecular fingerprints [43,54] at the most optimal K value (K = 3) and their neighboring K values. Each sample is represented by a single vertical line, proportional to the sample’s estimated ancestry of genetic components from *indica* or/and *japonica*. Many weedy rice samples showed the admixture of *indica* and *japonica* genetic components, particularly at K = 3 and K = 4.

**Figure 4 biology-12-00744-f004:**
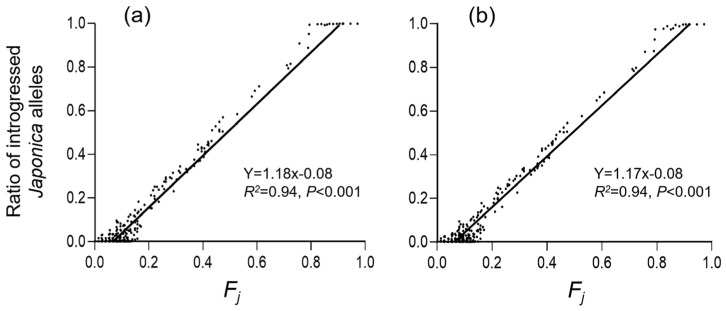
Correlation between the frequency of *japonica*-specific alleles (*F_j_*) based on the InDel molecular index [43] and the ratios of introgressed *japonica*-alleles (components) obtained from the STRUCTURE analysis in weedy rice populations from Jiangsu Province in China, when K = 2 (**a**) and K = 3 (**b**). Black dots represent weedy rice samples, black lines are regression lines.

**Figure 5 biology-12-00744-f005:**
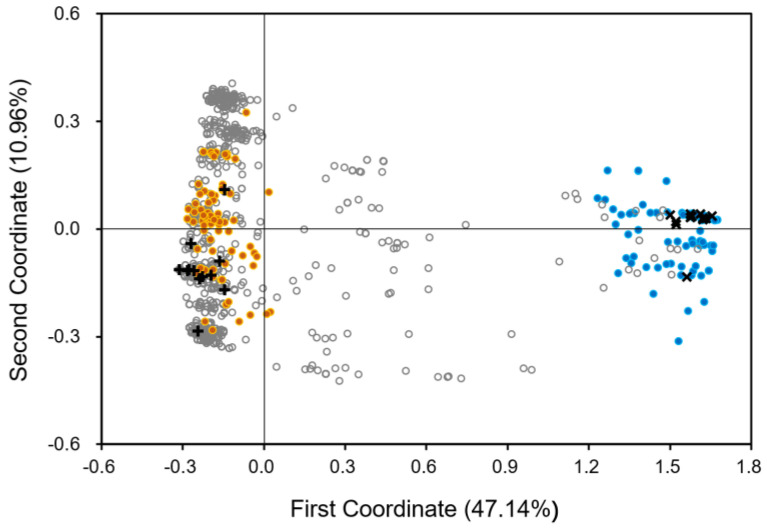
The scatterplot showing genetic relationships of the 1116 weedy rice samples from Jiangsu Province (JS), in addition to the reference samples including typical *indica* and *japonica* rice varieties, and weedy rice from Guangdong Province (GD) and northeast China (NEC) based on the principal coordinate analysis of the InDel dataset. Grey empty dots, JS weedy rice; +, typical *indica* rice varieties; ×, typical *japonica* rice varieties; orange dots, weedy rice from GD; blue dots, weedy rice from NEC.

**Figure 6 biology-12-00744-f006:**
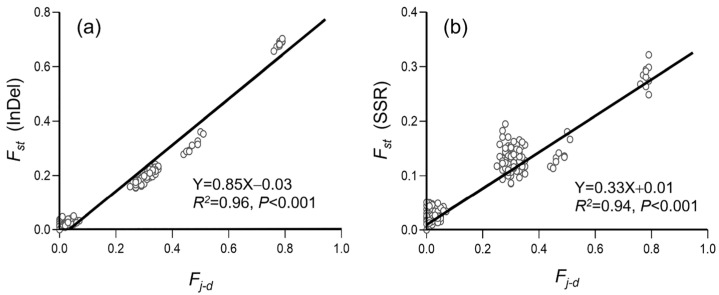
Correlation between the level of pairwise genetic differentiation (*F_st_*) and differences in crop-to-weed introgression (*F_j-d_*) of the ideally sampled weedy rice populations (empty dots) based on insertion/deletion (InDel, (**a**)) and simple sequence repeat (SSR, (**b**)) molecular fingerprints. Black lines are regression lines.

**Figure 7 biology-12-00744-f007:**
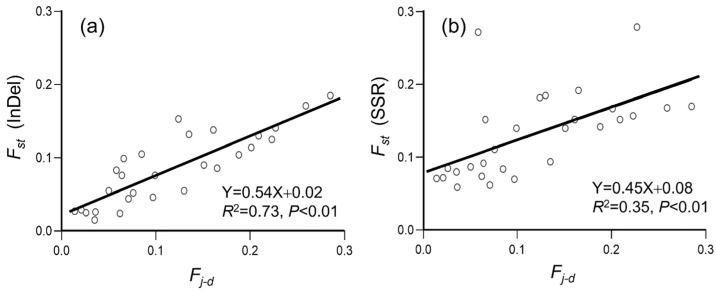
Correlation between the level of pairwise genetic differentiation (*F_st_*) and differences in crop-to-weed introgression (*F_j-d_*) of the naturally sampled weedy rice populations (empty dots), based on insertion/deletion (InDel, (**a**)) and simple sequence repeat (SSR, (**b**)) molecular fingerprints. Black lines are regression lines.

**Figure 8 biology-12-00744-f008:**
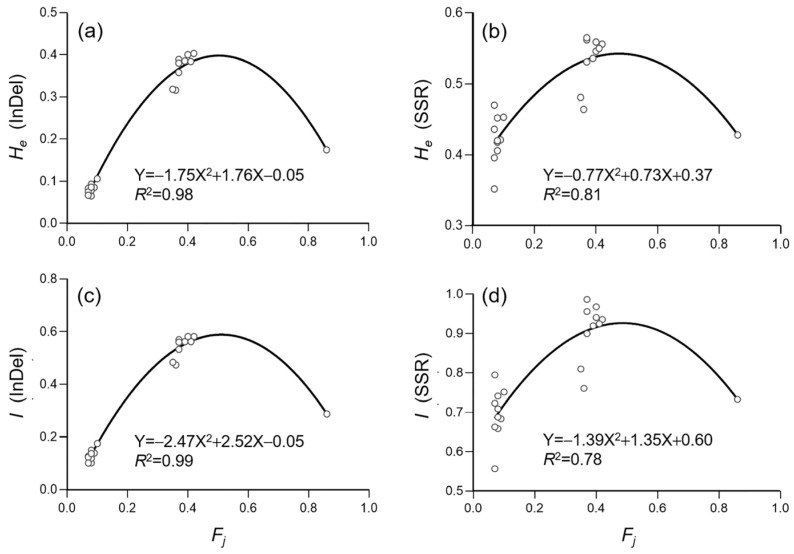
Results of regression between genetic diversity represented by Nei’s expected heterozygosity (*H_e_*) and Shannon’s information index *(I*) [59,60] and the frequency of *japonica*-specific alleles (*F_j_*) [43], based on the insertion/deletion (InDel, (**a**,**c**)) and simple sequence repeat (SSR, (**b**,**d**)) data matrices of the ideally sampled weedy rice populations (empty dots). Black curves are regression lines.

**Figure 9 biology-12-00744-f009:**
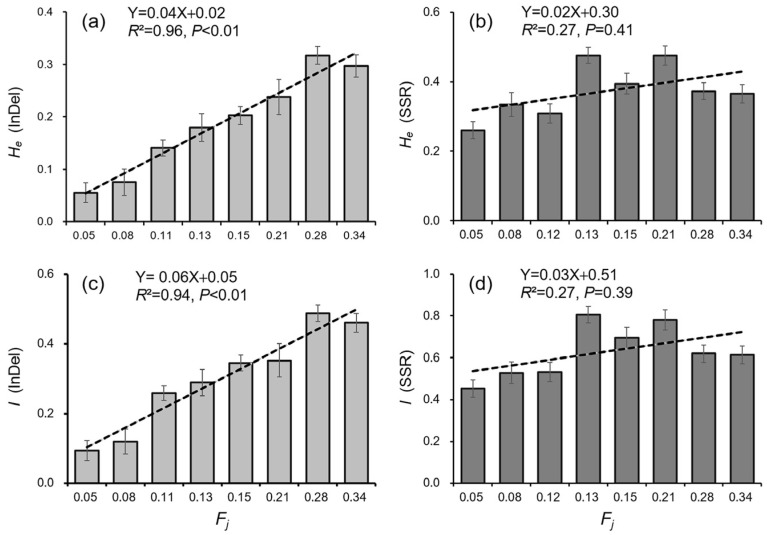
Correlations as represented by dashed lines (regression) between the genetic diversity parameters (*H_e_*, *I*) [59,60] and the *F_j_* values [43] of the eight naturally sampled weedy rice populations (NSWPs) from Jiangsu Province. The *F_j_* values and the genetic diversity parameters were calculated based on the data matrixes from the InDel (**a**,**c**) and SSR (**b**,**d**) molecular fingerprints, respectively. Each column represents an NSWP (n = 31), and the error bars indicate standard errors.

**Table 1 biology-12-00744-t001:** Identification of the *indica–japonica* characteristics of weedy rice samples from Guangdong and Jiangsu Provinces, and northeast China, based on the InDel molecular index. *F_j_* indicates the average values and standard deviation (in parenthesis).

Origin	*F_j_*	Type ^1^	Percent ^2^	Introgression ^3^
Guangdong	0.07 (0.021)	*Indica*	-	Low
Jiangsu	0.07 (0.001)	*Indica*	91.85%	Low
Jiangsu	0.38 (0.012)	Intermediate	6.72%	Middle
Jiangsu	0.86 (0.015)	*Japonica*	1.43%	High
Northeast	0.94 (0.058)	*Japonica*	-	Low

^1^ Classified based on the InDel index (*F_j_*) following the method of Lu et al. [43]; *F_j_* <0.25, *indica*; 0.25 ≤ *F_j_* < 0.75, intermediate; *japonica*, *F_j_* ≥ 0.75. ^2^ Only weedy rice samples from Jiangsu Province were counted. ^3^ Expected levels of *japonica* allelic introgression.

## Data Availability

Not applicable.

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
