# Peer review of "Crop–Weed Introgression Plays Critical Roles in Genetic Differentiation and Diversity of Weedy Rice: A Case Study of Human-Influenced Weed Evolution"

_biology, 2023, doi:10.3390/biology12050744_

Round 1
Reviewer 1 Report
This is an interesting study but not novel, concerning the crop-to-weed introgression on genetic differentiation of weedy rice in Jiangsu Province. Author used the InDel and SSR molecular fingerprints to analyze the allelic introgression from cultivated rice into weedy rice. The classification of indica-japonica characteristics for weedy rice in Jiangsu, based on the SSR and InDel molecular index was conducted and the genetic differentiation was shown. The results were same with the previous researches. But this study gives some evidences that human activities can influence the genetic differentiation and genetic diversity of weedy rice in the agroecosystems. In this perspective, the work gives us some useful information in future weed management strategy considering the crop variety improvement and replacement. However, there are some concerns that should be addressed before this work would be suitable for publication.
1. Materials and methods:
When were the weedy rice samples collected? What are the selection rules of typical indica and japonica types of weedy rice? It’s better to show the geographical distribution of weedy rice samples in Jiangsu visually in map.
2. Result
Figure 5, Figure 6 and Figure 7: What does the “*” in the fitting equations mean?
Figure 5, Figure 6: the captions should reflect the populations used and should not be identical; each graph in the combined figure should be numbered so as to present the results in the text clearly.
Figure 8: Does the level of genetic diversity significantly increase with the increase of Fj values? The significance analysis should be conducted. The curve fitting of the relationship between the genetic diversity and fj in natural weedy rice populations can be used as in other figures instead of bar charts in order to reflect the relationship more intuitively,
3. Reference
The format of references should be checked for consistency.
Delete the serial number “67”
4. Discussion
Is the change in rice cultivation entirely due to human behavior?
Do environmental factors play a role in the crop-weed Introgression form rice into weedy rice? For example, are there significant differences in the crop-weed Introgression of weedy rice collected from different latitudes?
Author Response
POINT-BY-POINT RESPONSES TO REVIEWERS’ COMMENTS
POINT-BY-POINT RESPONSES TO REVIEWER #1
General comments to the Authors
This is an interesting study but not novel, concerning the crop-to-weed introgression on genetic differentiation of weedy rice in Jiangsu Province. Author used the InDel and SSR molecular fingerprints to analyze the allelic introgression from cultivated rice into weedy rice. The classification of indica-japonica characteristics for weedy rice in Jiangsu, based on the SSR and InDel molecular index was conducted and the genetic differentiation was shown. The results were same with the previous researches. But this study gives some evidences that human activities can influence the genetic differentiation and genetic diversity of weedy rice in the agroecosystems. In this perspective, the work gives us some useful information in future weed management strategy considering the crop variety improvement and replacement. However, there are some concerns that should be addressed before this work would be suitable for publication.
Response: We appreciate these positive comments and shall revise the manuscript accordingly following the reviewer’s kind suggestions.
Comment-1 When were the weedy rice samples collected? What are the selection rules of typical indica and japonica types of weedy rice? It’s better to show the geographical distribution of weedy rice samples in Jiangsu visually in map.
Response: Weedy rice samples used in this study were collected in October, 2020~2021. As we already indicated in the manuscript, the determination of typical indica and japonica types of cultivated rice (not weedy rice) followed the previously published “InDel molecular index” (Lu et al. 2009, see reference 39). In this study, weedy rice was only determined as the indica (Fj <0.25), intermediate (0.25≤Fj <0.75), and japonica types (Fj≥0.75), respectively. Thanks for the comment, we added a distribution map of the sampling sites for weedy rice populations.
Comment-2 Figure 5, Figure 6 and Figure 7: What does the “*” in the fitting equations mean?
Response: Thanks for the comments. The “*” sign in the fitting equations indicates multiplication, which may (e.g., Figure 5-7) or may not (e.g., Figure 4) appear in the equations in different analytical software. For consistency, we deleted all these signs in Figure 5-7 in the revised manuscript.
Comment-3 Figure 5, Figure 6: the captions should reflect the populations used and should not be identical; each graph in the combined figure should be numbered so as to present the results in the text clearly.
Response: Thanks for the comment. However, the captions in Figure 5 and Figure 6 do not represent weedy rice populations, instead, the captions represent the correlation values between Fj-d and Fst of ideally-sampled weedy rice populations (Figure 5) and naturally-sampled weedy rice populations (Figure 6). Therefore, the correlation values cannot be indicated by population ID. Thanks for the last comment, we added (a) and (b) to differentiate each graph in the combined figure.
Comment-4 Figure 8: Does the level of genetic diversity significantly increase with the increase of Fj values? The significance analysis should be conducted. The curve fitting of the relationship between the genetic diversity and Fj in natural weedy rice populations can be used as in other figures instead of bar charts in order to reflect the relationship more intuitively,
Response: We appreciate the comments, and have added the results of correlation analysis between genetic diversity ​​(He, I) and the Fj values in Figure 8 by adding trend lines. In fact, the level of genetic diversity significantly increased with the increase of Fj values based on the InDel markers, but not significantly increased with the increase of Fj values based on the SSR markers. We have discussed this point in Discussion in the revised manuscript.
Comment-5 The format of references should be checked for consistency. Delete the serial number “67”
Response: Thank you for the comment and we have checked the format very carefully.
Comment-6 Is the change in rice cultivation entirely due to human behavior?
Response: Thanks for the comment. The crop cultivation including rice is determined by human, therefore, the change in rice cultivation is essentially influenced by human behavior, which has been mentioned in Introduction and Discussion of the manuscript.
Comment-7 Do environmental factors play a role in the crop-weed Introgression form rice into weedy rice? For example, are there significant differences in the crop-weed Introgression of weedy rice collected from different latitudes?
Response: Environmental factors, such as relative humidity, temperature, and wind speed, can play a role in the crop-weed introgression (e.g., Rong J, et al. Modelling pollen-mediated gene flow in rice: risk assessment and management of transgene escape. Plant Biotechnology Journal, 2010, 8:452-464). However, no evidence to show that the latitudes determine crop-weed introgression so far.
POINT-BY-POINT RESPONSES TO REVIEWER #2
General comments to the Authors
This manuscript is well-written and interesting, but it needs careful revision, as detailed below.
Line numbering is missing, and this makes reviewing less efficient.
Response: We indeed appreciate the positive comments from the reviewer. For the point of the line numbering, our original manuscript has the line numbers, the missing of the line numbers is probably due to the submission system.
Comment 1. Abstract: “Our findings provide strong evidence to support the hypothesis that introgression plays a critical role in shaping evolution of plant species”. On the one hand, I think it is not correct to extrapolate present findings about the case of crop-to-weed introgression in rice plants to evolution in general. In fact, “human activities such as the frequent change of crop varieties can strongly influence plant evolution in the agroecosystems”, which is not the case in most instances of evolution of plant species. On the other hand, it is a premise of this study that “introgression plays an essential role in shaping the evolution of plant species [5-13]”. So, using this premise to reach the same conclusion seems circular thinking. This is, rather, confirmation of a known effect in a special case.
Response: We appreciate the comments. We modified the sentences by softening the conclusion that “introgression plays a critical role in shaping evolution of plant species.” As an alternative, we have concentrated on the role of human influences in the evolution of weedy taxa, particularly in agroecosystems. We have modified relevant parts in the revised manuscript.
Comment 2. Page 2 “Obviously, introgression of japonica-specific alleles from japonica rice varieties into weedy rice with the indica genetic background has played an important role in changing indica-japonica characteristics of weedy rice in JS where the rapid change from planting indica to japonica rice varieties is taken place”. Unfortunately, this is not obvious at all: there is a strong reproductive barrier between indica and japonica subspecies, and hybrids between indica and japonica varieties are usually highly sterile (Chen et al. 2008. A triallelic system of S5 is a major regulator of the reproductive barrier and compatibility of indica–japonica hybrids in rice. Proc. Natl. Acad. Sci. USA 105(32):11436-11441; Zhu et al. 2017. Processes underlying a reproductive barrier in indica-japonica rice hybrids revealed by transcriptome analysis. Plant Physiol. 174(3):1683-1696). I personally found that a cross between two weedy genotypes from different origins (of which one was japonica, whereas it was not possible to establish the actual species of the other) failed to produce any viable seed. Although it is true that “When the cross-compatible plant species/populations come into contact, bidirectional or unidirectional introgression is likely to occur naturally through pollen-mediated gene flow [5,8]” (page 2), this is not the case between indica and japonica genotypes, unfortunately. So, it is more probable that changing indica-japonica characteristics of weedy rice following the rapid change from planting indica to japonica rice varieties was due to a few seeds of japonica weedy rice contaminating the seed of the imported japonica rice varieties. Unless, of course, the present Authors can hint to a mechanism overcoming inter-subspecific hybrid sterility (Guo et al. 2016. Overcoming inter-subspecific hybrid sterility in rice by developing indica-compatible japonica lines. Sci. Rep. 6:26878; Zhang 2022. The next generation of rice: inter-subspecific indica-japonica hybrid rice. Front. Plant Sci. 13) in their weedy rice populations. Please, also note: “HAS taken place”.
Response: Thank for the concerns. In fact, the differentiation of indica and japonica rice cultivars is a gradual process. The strong reproductive barriers present in the extreme cases (Chen et al. 2008; Zhu et al. 2017) where indica and japonica rice varieties cannot be successfully hybridized, but in many cases indica and japonica rice varieties can easily hybridize with a relatively high fertility. That is why there are many indica-japonica hybrid rice varieties in cultivation in China. Many research papers provided successful cases of indica-japonica hybrids in China and USA (Jia et al., 2012; Liu et al., 1996, 2023; Craig et al., 2014). In Jiangsu and many other provinces, indica and japonica rice (weedy rice) can hybridize with relatively high fertility.
Concerning the “seeds contamination” problem, we have no evidence to make such conclusion for Jiangsu rice cultivation, because agriculture in this province is highly developed, and most farmers use high-quality commercial “certified seeds” [see reference 32] in which the question of “seeds contamination” is not relevant. In addition, if the JS japonica type of weedy rice is from the “contaminated weedy rice seed” in JS rice fields, these weedy rice samples would not show admixture (caused by introgression) genetic components in the STRUCTURE analyses.
Comment 3. Three general routes have been hypothesized for the origin of weedy rice (reviewed in Ziska et al. 2015. Weedy (red) rice: an emerging constraint to global rice production. Adv. Agron. 129:181–228): (1) de-domestication of cultivated rice (namely, spontaneous mutation); (2) hybridization between wild rice species and cultivated groups; and (3) direct colonization of wild rice species of rice agricultural fields. Several studies have supported hypothesis 1 as more likely in determining weedy rice evolution in different world regions (Li et al. 2017. Signatures of adaptation in the weedy rice genome. Nature Genet. 49: 811-814; Qiu et al. 2017. Genomic variation associated with local adaptation of weedy rice during de-domestication. Nature Comm. 8:15323; Qiu et al. 2020. Diverse genetic mechanisms underlie worldwide convergent rice feralization. Genome Biol. 21:1-11). Quite interestingly, Ziska et al. (2015) say that hypothesis 1 is supported by the common observation that weedy rice is phenotypically “japonica-like” in regions where the japonica subspecies is predominantly cultivated, whereas in areas where the indica subspecies is dominant, weedy rice is more “indica-like”. However, the present Authors do not give much credit to hypothesis 1 in the case they describe (and I agree) given the quick change of sub-species alleles. This divergence of opinions must, however, be more clearly discussed. Other studies have documented hybridization between cultivated rice and wild O. rufipogon, or Oryza nivara, as a likely source of weedy rice origins, which supports hypothesis 2. Hypothesis 3 is poorly supported (Ziska et al. 2015). This ought to be better explained in the manuscript.
Response: Thanks for the comment. However, our manuscript does not study the origin of weedy rice in Jiangsu, therefore the questions proposed in this comment are not relevant to the manuscript.
Comment 4. Similarly, on page 5 it is said: “To estimate the possible application of the frequency of japonica-specific alleles (Fj) for measuring the levels of crop-to-weed introgression from japonica rice varieties”: this, in fact, assumes that introgression from japonica rice varieties, and not seed contamination, was the sole cause of variation in the frequency of japonica-specific alleles (Fj) of each weedy rice. As said, this assumption is not solid. The movement of weed seeds via crop contamination is an important mechanism of dispersal for certain weed species especially in direct-seeded rice systems in Asia (Rao et al. 2017. Preventive weed management in direct-seeded rice: targeting the weed seedbank. Adv. Agron. 144:45-142). Dispersal through seed contamination is particularly important for species like weedy rice, and it is a primary mechanism for this weed to invade rice fields (Rao et al. 2017). For example, in a survey in Vietnam, more than one-third of the collected rice seed samples were found contaminated with weedy rice seeds (Mai et al. 2000. Rice seed contamination in Vietnam. Pp. 17-19 in: Wild and Weedy Rice in Rice Ecosystems in Asia: A Review. Baki, Chin, Mortimer, Eds. Limited Proceedings No. 2, IRRI, Los Baños, Philippines).
Response: Thank you for the comment. Please see our responses to comment 2 about this point. Commonly, the question of “seed contamination” is related to the uses of self-saved seeds for cultivation.
Comment 5. The seed contamination hypothesis could very well explain the “small proportion of weedy rice samples from JS” that “were scattered among the typical japonica rice varieties … (right in Figure 4)” (page 9). If these weedy rices were originated from the cross between the japonica crop and the old indica weedy rices, the japonica weedy rices should start in the middle (of genetic diversity) between indica and japonica rice varieties and from there they should shift toward the japonica varieties (because of convergence adaptation) without fully mix with them. The existence of weedy rices both closer to the indica crops and intermixed with the japonica crop rices seems harder to explain with a model based exclusively on the introgression from japonica rice cultivars into the indica type of weedy rice.
Response: Thank you for the comment. Please see our responses to comment 2 about this point.
Comment 6. Although it may be true that “This finding was supported by the gradually increased japonica-specific allelic frequency (Fj) in the JS weedy rice samples that should originally be the indica type” (page 10), the same could have been observed if japonica weedy rices, introduced together with the japonica cultivars, had progressively crossed with the old indica weedy rices. That is, the direction of the change of weedy rice in Figure 4 could be from the right to the left instead of from left to right (at least for the swam at the positive loads of the first principal coordinate). The simulation performed with ideal weedy rice populations in Figure 5 does not negate that a phenomenon different from crop allelic introgression could cause “genetic differentiation of its conspecific weed” as well.
Response: Concerning the point about “seed contamination” by introduced japonica weedy rice, please see our responses to comment 2.
Reviewer’s point about “japonica cultivars, had progressively crossed with the old indica weedy rices” directly rules out reviewer’s comment 2 (“there is a strong reproductive barrier between indica and japonica subspecies, and hybrids between indica and japonica varieties are usually highly sterile”), and therefore, we cannot have correct answers to these contradictory questions.
The question “The simulation performed with ideal weedy rice populations in Figure 5 does not negate that a phenomenon different from crop allelic introgression could cause “genetic differentiation of its conspecific weed” as well” is not so clear. But, Figure 5 clearly indicates the correlation of allelic introgression with genetic differentiation based on the results from Figure 3.
Comment 7. The seed contamination hypothesis would also explain pretty well why “In this study, we also found that the reference weedy rice populations from NEC were essentially the japonica type that was associated strongly with the typical japonica rice varieties; whereas those from GD were essentially the indica type that was associated intimately with the typical indica rice varieties” (page 13).
Response: Concerning the question of “seed contamination”, please see our responses to comment 2. The reason why “the reference weedy rice populations from NEC were essentially the japonica type… those from GD were essentially the indica type” is because rice cultivars in NEC nor in GD did not experience the dramatic cultivation change between indica and japonica rice varieties. Therefore, some weedy rice populations in Jiangsu showed intermediate types through indica-japonica introgression.
Comment 8. Moreover, this whole study, and particularly section 2.4.1 (“Indica-japonica characterization of weedy rice and cultivated rice”), assumes that the weedy rices were Oryza sativa, either indica, japonica, or intermediate. Although weedy rice is often classified as Oryza sativa, other Oryza species such as Oryza rufipogon, Oryza barthii, Oryza nivara, and Oryza longistaminata have also been proposed as sources of weedy rice (Ziska et al. 2015); since, as mentioned above, hypothesis 2 (namely, hybridization between wild rice species and the cultivated crop) appears probable in several cases. Specifically, in regions of tropical Asia where rice is grown in proximity to its wild progenitor (Oryza rufipogon), gene flow from wild populations also contributes to the genetic composition of weed populations ([27,28]; Pusadee et al. 2013. Population structure of the primary gene pool of Oryza sativa in Thailand. Genet. Resour. Crop Evol. 60:335–353; Wang et al. 2017. Asian wild rice is a hybrid swarm with extensive gene flow and feralization from domesticated rice. Genome Res. 27:1029-1038). In general, weedy rice is a complex of taxonomically not well defined Oryza species, hybrids and special biotypes (Kraehmer et al. 2016. Global distribution of rice weeds–a review. Crop Protection 80:73-86). So, how did the present Authors account for the eventuality that portions of the weedy rice populations genome can origin from species different from Oryza sativa?
Response: Concerning the point about the origin of weedy rice, please see our responses to comment 3.
Comment 9. what measure unit is “khm”?
Response: Thanks for the comment. We have corrected the error and replaced "khm2" with "1000 ha" (1000 hectares, kha) in the revised manuscript.
Comment 10. Table 1: what are the number within parenthesis in the second column?
Response: As we already indicated in the table caption of Table 1 that the numbers within parentheses in the second column were the standard deviation of the Fj values.
Comment 11. Page 5: maybe I am a bit simple-minded, but I do not understand the difference between “the Fj values and the proportion of introgressed japonica genetic components (alleles)”. Could the Authors explain what is the material difference between these two indices and why they should, or could, diverge? This would greatly help in understanding why a good correlation between them (Figure 3) suggests “that the Fj values … could be used to estimate … the relationships of crop-to-weed introgression with genetic differentiation and genetic diversity”. This whole sentence appears to be trying to convey an important conceptual point, but I do not understand it. Some more explanation would be really appreciated by the general readership.
Response: Thanks for the questions. Actually, the Fj value and the ratio of introgressed japonica alleles are different. The Fj value is the average frequency of the japonica-specific alleles calculated by the following formula (Xjj indicates the homozygous japonica genotype, Xij indicates the heterozygous indica–japonica genotype) [39].
The ratio of introgressed japonica alleles is the proportion of membership of japonica population, which is calculated with InDel genotyping data matrix by STRUCTURE population Q-matrix [50]. The ratio of introgressed japonica alleles represents the level of introgression. Only when Fj value is proved to be positively correlated with the ratio, the Fj values can be used directly to estimate the level of introgression.
Comment 12. Page 8: change “most JS samples weedy rice” to “most JS weedy rice samples”.
Response: Thanks a lot for the comment. We have changed the use of "most JS samples weedy rice" to "most JS weedy rice samples" in the revised manuscript.
Comment 13. Page 8, “the typical indica rice reference varieties showed somehow other genetic components”: I do not know about China indica cultivars, but most Western indica-type cultivars are, in fact, derived from crosses between indica and japonica rices (a few that were obtained years ago by using bridge genotypes), aimed at transferring the indica-type grain in a genetic background more suitable for temperate regions.
Response: Thanks for the comments. Chinese traditional rice varieties generally adopt systematic selection, with a relatively clear genetic background. Meanwhile, we also emphasized in the manuscript that“the typical japonica rice reference varieties showed distinctly unique genetic component (green) with nearly no admixture, although the typical indica rice reference varieties showed somehow other genetic components”. Indica varieties showing other genetic components only indicate indica varieties have greater genetic diversity, which has no influence on the conclusions in this study.
Comment 14. As for the results described on page 11 and Figure 8, it is not clear whether weedy samples with Fj >0.5 matched the expected, ideal trend described in Figure 7.
Response: Thanks a lot for this comment. Many studies indicate that weedy rice in JS is mainly indica type. Due to the self-pollination feature of cultivated rice and weedy rice, there are relatively few intermediate and japonica type of weedy rice in the naturally-sampled weedy rice populations. With only a few decades of crop-to-weed introgression from japonica rice varieties, it is natural that no naturally sampled populations had Fj >0.5.
Comment 15. Figure 8 also shows that InDel are better to highlight the relationship between the level of genetic diversity and crop-to-weed introgression than SSR: could the Authors elaborate a bit on this?
Response: Thanks a lot for this comment. The differences between genetic diversity revealed by SSR and InDel molecular markers can easily be explained by the reasons that the formation of genetic diversity in weedy rice is not only determined by crop-weed introgression involving indica-japonica alleles, but also by other types of alleles that are not associated with the indica and japonica characteristics. We already discussed the differences in the Discussion section of our manuscript.
Comment 16. Page 14: Isn’t the fact that “introgression … can … cause … genetic differentiation” well-known? It is certainly important to note that it was observed in this particular case too, but a suitable reference could be introduced to remind that this is, indeed, a general expectation. Specifically, that genetic introgression between cultivated and weedy plants caused high diversity of weedy red rice has already been reported, for example, by Shivrain et al. (2010. Diversity of weedy red rice (Oryza sativa L.) in Arkansas, USA in relation to weed management. Crop Protection 29:721-730 and references wherein).
Response: Thanks for this comment. The detail mechanism and process of introgression that influences genetic differentiation are not completely known yet. Our point is to use concrete event to explain how increased crop-to-weed introgression can influence the genetic differentiation of weedy rice populations, particularly under human influences. However, the reference provided by the reviewer mainly involves the study of morphological variability in weedy rice, which is different from our findings about genetic differentiation.
Comment 17. Throughout the manuscript, the Authors sometimes refer to old (indica) weedy rices populations as “natural weedy rice populations”: as weedy rice is typical of human agrosystems, it seems a bit forced to consider them as “natural”.
Response: Thanks a lot for the comment. We have modiffied our definition for “ideally-sampled weedy rice population, ISWP” and “naturally-sampled weedy rice population, NSWP”, and used these terminologies consistently throughout the revised manuscript.
Comment 18. Finally, can the present Authors suggest a relationship between the weedy rice populations they studied in this work and the biotypes described by Wang et al. (2023. Occurrence pattern and morphological polymorphism of Chinese weedy rice. J. Integr. Agric. 22:149-169) for the same regions?
Response: Thanks for the comment. However, after carefully reading Wang’s study, we found that the authors classified Chinese weedy rice into three major groups by morphology: multi-tiller (group1), large-leaf (group 2), and large-seed (group 3) weedy rice. Our study is essentially focused on genetic differentiation and diversity of weedy rice from Jiangsu using molecular fingerprints, which is not relevant to Wang’s study.
Comment 19. All in all, the present manuscript presents an interesting case study, with many important data, but it fails to consider important facts (namely, the origin of weedy rices from other species) and alternative hypotheses (that is, weedy rice populations of a different sub-species can be imported as contaminants of the crop seed). I believe, anyway, that the Authors can dutifully account for the missing pieces of information by extensively, and carefully, revise the manuscript.
Response: We indeed appreciate the reviewer’s constructive comments and suggestions. We have carefully revised the manuscript by concentrating more on the results of human influenced weed evolution in agroecosystems.

Reviewer 2 Report
This manuscript is well-written and interesting, but it needs careful revision, as detailed below.
Line numbering is missing, and this makes reviewing less efficient.
Abstract: “Our findings provide strong evidence to support the hypothesis that introgression plays a critical role in shaping evolution of plant species”. On the one hand, I think it is not correct to extrapolate present findings about the case of crop-to-weed introgression in rice plants to evolution in general. In fact, “human activities such as the frequent change of crop varieties can strongly influence plant evolution in the agroecosystems”, which is not the case in most instances of evolution of plant species. On the other hand, it is a premise of this study that “introgression plays an essential role in shaping the evolution of plant species [5-13]”. So, using this premise to reach the same conclusion seems circular thinking. This is, rather, confirmation of a known effect in a special case.
Page 2 “Obviously, introgression of japonica-specific alleles from japonica rice varieties into weedy rice with the indica genetic background has played an important role in changing indica-japonica characteristics of weedy rice in JS where the rapid change from planting indica to japonica rice varieties is taken place”. Unfortunately, this is not obvious at all: there is a strong reproductive barrier between indica and japonica subspecies, and hybrids between indica and japonica varieties are usually highly sterile (Chen et al. 2008. A triallelic system of S5 is a major regulator of the reproductive barrier and compatibility of indica–japonica hybrids in rice. Proc. Natl. Acad. Sci. USA 105(32):11436-11441; Zhu et al. 2017. Processes underlying a reproductive barrier in indica-japonica rice hybrids revealed by transcriptome analysis. Plant Physiol. 174(3):1683-1696). I personally found that a cross between two weedy genotypes from different origins (of which one was japonica, whereas it was not possible to establish the actual species of the other) failed to produce any viable seed. Although it is true that “When the cross-compatible plant species/populations come into contact, bidirectional or unidirectional introgression is likely to occur naturally through pollen-mediated gene flow [5,8]” (page 2), this is not the case between indica and japonica genotypes, unfortunately. So, it is more probable that changing indica-japonica characteristics of weedy rice following the rapid change from planting indica to japonica rice varieties was due to a few seeds of japonica weedy rice contaminating the seed of the imported japonica rice varieties. Unless, of course, the present Authors can hint to a mechanism overcoming inter-subspecific hybrid sterility (Guo et al. 2016. Overcoming inter-subspecific hybrid sterility in rice by developing indica-compatible japonica lines. Sci. Rep. 6:26878; Zhang 2022. The next generation of rice: inter-subspecific indica-japonica hybrid rice. Front. Plant Sci. 13) in their weedy rice populations. Please, also note: “HAS taken place”.
Three general routes have been hypothesized for the origin of weedy rice (reviewed in Ziska et al. 2015. Weedy (red) rice: an emerging constraint to global rice production. Adv. Agron. 129:181–228): (1) de-domestication of cultivated rice (namely, spontaneous mutation); (2) hybridization between wild rice species and cultivated groups; and (3) direct colonization of wild rice species of rice agricultural fields. Several studies have supported hypothesis 1 as more likely in determining weedy rice evolution in different world regions (Li et al. 2017. Signatures of adaptation in the weedy rice genome. Nature Genet. 49: 811-814; Qiu et al. 2017. Genomic variation associated with local adaptation of weedy rice during de-domestication. Nature Comm. 8:15323; Qiu et al. 2020. Diverse genetic mechanisms underlie worldwide convergent rice feralization. Genome Biol. 21:1-11). Quite interestingly, Ziska et al. (2015) say that hypothesis 1 is supported by the common observation that weedy rice is phenotypically “japonica-like” in regions where the japonica subspecies is predominantly cultivated, whereas in areas where the indica subspecies is dominant, weedy rice is more “indica-like”. However, the present Authors do not give much credit to hypothesis 1 in the case they describe (and I agree) given the quick change of sub-species alleles. This divergence of opinions must, however, be more clearly discussed. Other studies have documented hybridization between cultivated rice and wild O. rufipogon, or Oryza nivara, as a likely source of weedy rice origins, which supports hypothesis 2. Hypothesis 3 is poorly supported (Ziska et al. 2015). This ought to be better explained in the manuscript.
Similarly, on page 5 it is said: “To estimate the possible application of the frequency of japonica-specific alleles (Fj) for measuring the levels of crop-to-weed introgression from japonica rice varieties”: this, in fact, assumes that introgression from japonica rice varieties, and not seed contamination, was the sole cause of variation in the frequency of japonica-specific alleles (Fj) of each weedy rice. As said, this assumption is not solid. The movement of weed seeds via crop contamination is an important mechanism of dispersal for certain weed species especially in direct-seeded rice systems in Asia (Rao et al. 2017. Preventive weed management in direct-seeded rice: targeting the weed seedbank. Adv. Agron. 144:45-142). Dispersal through seed contamination is particularly important for species like weedy rice, and it is a primary mechanism for this weed to invade rice fields (Rao et al. 2017). For example, in a survey in Vietnam, more than one-third of the collected rice seed samples were found contaminated with weedy rice seeds (Mai et al. 2000. Rice seed contamination in Vietnam. Pp. 17-19 in: Wild and Weedy Rice in Rice Ecosystems in Asia: A Review. Baki, Chin, Mortimer, Eds. Limited Proceedings No. 2, IRRI, Los Baños, Philippines).
The seed contamination hypothesis could very well explain the “small proportion of weedy rice samples from JS” that “were scattered among the typical japonica rice varieties … (right in Figure 4)” (page 9). If these weedy rices were originated from the cross between the japonica crop and the old indica weedy rices, the japonica weedy rices should start in the middle (of genetic diversity) between indica and japonica rice varieties and from there they should shift toward the japonica varieties (because of convergence adaptation) without fully mix with them. The existence of weedy rices both closer to the indica crops and intermixed with the japonica crop rices seems harder to explain with a model based exclusively on the introgression from japonica rice cultivars into the indica type of weedy rice.
Although it may be true that “This finding was supported by the gradually increased japonica-specific allelic frequency (Fj) in the JS weedy rice samples that should originally be the indica type” (page 10), the same could have been observed if japonica weedy rices, introduced together with the japonica cultivars, had progressively crossed with the old indica weedy rices. That is, the direction of the change of weedy rice in Figure 4 could be from the right to the left instead of from left to right (at least for the swam at the positive loads of the first principal coordinate). The simulation performed with ideal weedy rice populations in Figure 5 does not negate that a phenomenon different from crop allelic introgression could cause “genetic differentiation of its conspecific weed” as well.
The seed contamination hypothesis would also explain pretty well why “In this study, we also found that the reference weedy rice populations from NEC were essentially the japonica type that was associated strongly with the typical japonica rice varieties; whereas those from GD were essentially the indica type that was associated intimately with the typical indica rice varieties” (page 13).
Moreover, this whole study, and particularly section 2.4.1 (“Indica-japonica characterization of weedy rice and cultivated rice”), assumes that the weedy rices were Oryza sativa, either indica, japonica, or intermediate. Although weedy rice is often classified as Oryza sativa, other Oryza species such as Oryza rufipogon, Oryza barthii, Oryza nivara, and Oryza longistaminata have also been proposed as sources of weedy rice (Ziska et al. 2015); since, as mentioned above, hypothesis 2 (namely, hybridization between wild rice species and the cultivated crop) appears probable in several cases. Specifically, in regions of tropical Asia where rice is grown in proximity to its wild progenitor (Oryza rufipogon), gene flow from wild populations also contributes to the genetic composition of weed populations ([27,28]; Pusadee et al. 2013. Population structure of the primary gene pool of Oryza sativa in Thailand. Genet. Resour. Crop Evol. 60:335–353; Wang et al. 2017. Asian wild rice is a hybrid swarm with extensive gene flow and feralization from domesticated rice. Genome Res. 27:1029-1038). In general, weedy rice is a complex of taxonomically not well defined Oryza species, hybrids and special biotypes (Kraehmer et al. 2016. Global distribution of rice weeds–a review. Crop Protection 80:73-86). So, how did the present Authors account for the eventuality that portions of the weedy rice populations genome can origin from species different from Oryza sativa?
Page 3: what measure unit is “khm2”?
Table 1: what are the number within parenthesis in the second column?
Page 5: maybe I am a bit simple-minded, but I do not understand the difference between “the Fj values and the proportion of introgressed japonica genetic components (alleles)”. Could the Authors explain what is the material difference between these two indices and why they should, or could, diverge? This would greatly help in understanding why a good correlation between them (Figure 3) suggests “that the Fj values … could be used to estimate … the relationships of crop-to-weed introgression with genetic differentiation and genetic diversity”. This whole sentence appears to be trying to convey an important conceptual point, but I do not understand it. Some more explanation would be really appreciated by the general readership.
Page 8: change “most JS samples weedy rice” to “most JS weedy rice samples”.
Page 8, “the typical indica rice reference varieties showed somehow other genetic components”: I do not know about China indica cultivars, but most Western indica-type cultivars are, in fact, derived from crosses between indica and japonica rices (a few that were obtained years ago by using bridge genotypes), aimed at transferring the indica-type grain in a genetic background more suitable for temperate regions.
As for the results described on page 11 and Figure 8, it is not clear whether weedy samples with Fj >0.5 matched the expected, ideal trend described in Figure 7.
Figure 8 also shows that InDel are better to highlight the relationship between the level of genetic diversity and crop-to-weed introgression than SSR: could the Authors elaborate a bit on this?
Page 14: Isn’t the fact that “introgression … can … cause … genetic differentiation” well-known? It is certainly important to note that it was observed in this particular case too, but a suitable reference could be introduced to remind that this is, indeed, a general expectation. Specifically, that genetic introgression between cultivated and weedy plants caused high diversity of weedy red rice has already been reported, for example, by Shivrain et al. (2010. Diversity of weedy red rice (Oryza sativa L.) in Arkansas, USA in relation to weed management. Crop Protection 29:721-730 and references wherein).
Throughout the manuscript, the Authors sometimes refer to old (indica) weedy rices populations as “natural weedy rice populations”: as weedy rice is typical of human agrosystems, it seems a bit forced to consider them as “natural”.
Finally, can the present Authors suggest a relationship between the weedy rice populations they studied in this work and the biotypes described by Wang et al. (2023. Occurrence pattern and morphological polymorphism of Chinese weedy rice. J. Integr. Agric. 22:149-169) for the same regions?
All in all, the present manuscript presents an interesting case study, with many important data, but it fails to consider important facts (namely, the origin of weedy rices from other species) and alternative hypotheses (that is, weedy rice populations of a different sub-species can be imported as contaminants of the crop seed). I believe, anyway, that the Authors can dutifully account for the missing pieces of information by extensively, and carefully, revise the manuscript.
Author Response
POINT-BY-POINT RESPONSES TO REVIEWER #2
General comments to the Authors
This manuscript is well-written and interesting, but it needs careful revision, as detailed below.
Line numbering is missing, and this makes reviewing less efficient.
Response: We indeed appreciate the positive comments from the reviewer. For the point of the line numbering, our original manuscript has the line numbers, the missing of the line numbers is probably due to the submission system.
Comment 1. Abstract: “Our findings provide strong evidence to support the hypothesis that introgression plays a critical role in shaping evolution of plant species”. On the one hand, I think it is not correct to extrapolate present findings about the case of crop-to-weed introgression in rice plants to evolution in general. In fact, “human activities such as the frequent change of crop varieties can strongly influence plant evolution in the agroecosystems”, which is not the case in most instances of evolution of plant species. On the other hand, it is a premise of this study that “introgression plays an essential role in shaping the evolution of plant species [5-13]”. So, using this premise to reach the same conclusion seems circular thinking. This is, rather, confirmation of a known effect in a special case.
Response: We appreciate the comments. We modified the sentences by softening the conclusion that “introgression plays a critical role in shaping evolution of plant species.” As an alternative, we have concentrated on the role of human influences in the evolution of weedy taxa, particularly in agroecosystems. We have modified relevant parts in the revised manuscript.
Comment 2. Page 2 “Obviously, introgression of japonica-specific alleles from japonica rice varieties into weedy rice with the indica genetic background has played an important role in changing indica-japonica characteristics of weedy rice in JS where the rapid change from planting indica to japonica rice varieties is taken place”. Unfortunately, this is not obvious at all: there is a strong reproductive barrier between indica and japonica subspecies, and hybrids between indica and japonica varieties are usually highly sterile (Chen et al. 2008. A triallelic system of S5 is a major regulator of the reproductive barrier and compatibility of indica–japonica hybrids in rice. Proc. Natl. Acad. Sci. USA 105(32):11436-11441; Zhu et al. 2017. Processes underlying a reproductive barrier in indica-japonica rice hybrids revealed by transcriptome analysis. Plant Physiol. 174(3):1683-1696). I personally found that a cross between two weedy genotypes from different origins (of which one was japonica, whereas it was not possible to establish the actual species of the other) failed to produce any viable seed. Although it is true that “When the cross-compatible plant species/populations come into contact, bidirectional or unidirectional introgression is likely to occur naturally through pollen-mediated gene flow [5,8]” (page 2), this is not the case between indica and japonica genotypes, unfortunately. So, it is more probable that changing indica-japonica characteristics of weedy rice following the rapid change from planting indica to japonica rice varieties was due to a few seeds of japonica weedy rice contaminating the seed of the imported japonica rice varieties. Unless, of course, the present Authors can hint to a mechanism overcoming inter-subspecific hybrid sterility (Guo et al. 2016. Overcoming inter-subspecific hybrid sterility in rice by developing indica-compatible japonica lines. Sci. Rep. 6:26878; Zhang 2022. The next generation of rice: inter-subspecific indica-japonica hybrid rice. Front. Plant Sci. 13) in their weedy rice populations. Please, also note: “HAS taken place”.
Response: Thank for the concerns. In fact, the differentiation of indica and japonica rice cultivars is a gradual process. The strong reproductive barriers present in the extreme cases (Chen et al. 2008; Zhu et al. 2017) where indica and japonica rice varieties cannot be successfully hybridized, but in many cases indica and japonica rice varieties can easily hybridize with a relatively high fertility. That is why there are many indica-japonica hybrid rice varieties in cultivation in China. Many research papers provided successful cases of indica-japonica hybrids in China and USA (Jia et al., 2012; Liu et al., 1996, 2023; Craig et al., 2014). In Jiangsu and many other provinces, indica and japonica rice (weedy rice) can hybridize with relatively high fertility.
Concerning the “seeds contamination” problem, we have no evidence to make such conclusion for Jiangsu rice cultivation, because agriculture in this province is highly developed, and most farmers use high-quality commercial “certified seeds” [see reference 32] in which the question of “seeds contamination” is not relevant. In addition, if the JS japonica type of weedy rice is from the “contaminated weedy rice seed” in JS rice fields, these weedy rice samples would not show admixture (caused by introgression) genetic components in the STRUCTURE analyses.
Comment 3. Three general routes have been hypothesized for the origin of weedy rice (reviewed in Ziska et al. 2015. Weedy (red) rice: an emerging constraint to global rice production. Adv. Agron. 129:181–228): (1) de-domestication of cultivated rice (namely, spontaneous mutation); (2) hybridization between wild rice species and cultivated groups; and (3) direct colonization of wild rice species of rice agricultural fields. Several studies have supported hypothesis 1 as more likely in determining weedy rice evolution in different world regions (Li et al. 2017. Signatures of adaptation in the weedy rice genome. Nature Genet. 49: 811-814; Qiu et al. 2017. Genomic variation associated with local adaptation of weedy rice during de-domestication. Nature Comm. 8:15323; Qiu et al. 2020. Diverse genetic mechanisms underlie worldwide convergent rice feralization. Genome Biol. 21:1-11). Quite interestingly, Ziska et al. (2015) say that hypothesis 1 is supported by the common observation that weedy rice is phenotypically “japonica-like” in regions where the japonica subspecies is predominantly cultivated, whereas in areas where the indica subspecies is dominant, weedy rice is more “indica-like”. However, the present Authors do not give much credit to hypothesis 1 in the case they describe (and I agree) given the quick change of sub-species alleles. This divergence of opinions must, however, be more clearly discussed. Other studies have documented hybridization between cultivated rice and wild O. rufipogon, or Oryza nivara, as a likely source of weedy rice origins, which supports hypothesis 2. Hypothesis 3 is poorly supported (Ziska et al. 2015). This ought to be better explained in the manuscript.
Response: Thanks for the comment. However, our manuscript does not study the origin of weedy rice in Jiangsu, therefore the questions proposed in this comment are not relevant to the manuscript.
Comment 4. Similarly, on page 5 it is said: “To estimate the possible application of the frequency of japonica-specific alleles (Fj) for measuring the levels of crop-to-weed introgression from japonica rice varieties”: this, in fact, assumes that introgression from japonica rice varieties, and not seed contamination, was the sole cause of variation in the frequency of japonica-specific alleles (Fj) of each weedy rice. As said, this assumption is not solid. The movement of weed seeds via crop contamination is an important mechanism of dispersal for certain weed species especially in direct-seeded rice systems in Asia (Rao et al. 2017. Preventive weed management in direct-seeded rice: targeting the weed seedbank. Adv. Agron. 144:45-142). Dispersal through seed contamination is particularly important for species like weedy rice, and it is a primary mechanism for this weed to invade rice fields (Rao et al. 2017). For example, in a survey in Vietnam, more than one-third of the collected rice seed samples were found contaminated with weedy rice seeds (Mai et al. 2000. Rice seed contamination in Vietnam. Pp. 17-19 in: Wild and Weedy Rice in Rice Ecosystems in Asia: A Review. Baki, Chin, Mortimer, Eds. Limited Proceedings No. 2, IRRI, Los Baños, Philippines).
Response: Thank you for the comment. Please see our responses to comment 2 about this point. Commonly, the question of “seed contamination” is related to the uses of self-saved seeds for cultivation.
Comment 5. The seed contamination hypothesis could very well explain the “small proportion of weedy rice samples from JS” that “were scattered among the typical japonica rice varieties … (right in Figure 4)” (page 9). If these weedy rices were originated from the cross between the japonica crop and the old indica weedy rices, the japonica weedy rices should start in the middle (of genetic diversity) between indica and japonica rice varieties and from there they should shift toward the japonica varieties (because of convergence adaptation) without fully mix with them. The existence of weedy rices both closer to the indica crops and intermixed with the japonica crop rices seems harder to explain with a model based exclusively on the introgression from japonica rice cultivars into the indica type of weedy rice.
Response: Thank you for the comment. Please see our responses to comment 2 about this point.
Comment 6. Although it may be true that “This finding was supported by the gradually increased japonica-specific allelic frequency (Fj) in the JS weedy rice samples that should originally be the indica type” (page 10), the same could have been observed if japonica weedy rices, introduced together with the japonica cultivars, had progressively crossed with the old indica weedy rices. That is, the direction of the change of weedy rice in Figure 4 could be from the right to the left instead of from left to right (at least for the swam at the positive loads of the first principal coordinate). The simulation performed with ideal weedy rice populations in Figure 5 does not negate that a phenomenon different from crop allelic introgression could cause “genetic differentiation of its conspecific weed” as well.
Response: Concerning the point about “seed contamination” by introduced japonica weedy rice, please see our responses to comment 2.
Reviewer’s point about “japonica cultivars, had progressively crossed with the old indica weedy rices” directly rules out reviewer’s comment 2 (“there is a strong reproductive barrier between indica and japonica subspecies, and hybrids between indica and japonica varieties are usually highly sterile”), and therefore, we cannot have correct answers to these contradictory questions.
The question “The simulation performed with ideal weedy rice populations in Figure 5 does not negate that a phenomenon different from crop allelic introgression could cause “genetic differentiation of its conspecific weed” as well” is not so clear. But, Figure 5 clearly indicates the correlation of allelic introgression with genetic differentiation based on the results from Figure 3.
Comment 7. The seed contamination hypothesis would also explain pretty well why “In this study, we also found that the reference weedy rice populations from NEC were essentially the japonica type that was associated strongly with the typical japonica rice varieties; whereas those from GD were essentially the indica type that was associated intimately with the typical indica rice varieties” (page 13).
Response: Concerning the question of “seed contamination”, please see our responses to comment 2. The reason why “the reference weedy rice populations from NEC were essentially the japonica type… those from GD were essentially the indica type” is because rice cultivars in NEC nor in GD did not experience the dramatic cultivation change between indica and japonica rice varieties. Therefore, some weedy rice populations in Jiangsu showed intermediate types through indica-japonica introgression.
Comment 8. Moreover, this whole study, and particularly section 2.4.1 (“Indica-japonica characterization of weedy rice and cultivated rice”), assumes that the weedy rices were Oryza sativa, either indica, japonica, or intermediate. Although weedy rice is often classified as Oryza sativa, other Oryza species such as Oryza rufipogon, Oryza barthii, Oryza nivara, and Oryza longistaminata have also been proposed as sources of weedy rice (Ziska et al. 2015); since, as mentioned above, hypothesis 2 (namely, hybridization between wild rice species and the cultivated crop) appears probable in several cases. Specifically, in regions of tropical Asia where rice is grown in proximity to its wild progenitor (Oryza rufipogon), gene flow from wild populations also contributes to the genetic composition of weed populations ([27,28]; Pusadee et al. 2013. Population structure of the primary gene pool of Oryza sativa in Thailand. Genet. Resour. Crop Evol. 60:335–353; Wang et al. 2017. Asian wild rice is a hybrid swarm with extensive gene flow and feralization from domesticated rice. Genome Res. 27:1029-1038). In general, weedy rice is a complex of taxonomically not well defined Oryza species, hybrids and special biotypes (Kraehmer et al. 2016. Global distribution of rice weeds–a review. Crop Protection 80:73-86). So, how did the present Authors account for the eventuality that portions of the weedy rice populations genome can origin from species different from Oryza sativa?
Response: Concerning the point about the origin of weedy rice, please see our responses to comment 3.
Comment 9. what measure unit is “khm”?
Response: Thanks for the comment. We have corrected the error and replaced "khm2" with "1000 ha" (1000 hectares, kha) in the revised manuscript.
Comment 10. Table 1: what are the number within parenthesis in the second column?
Response: As we already indicated in the table caption of Table 1 that the numbers within parentheses in the second column were the standard deviation of the Fj values.
Comment 11. Page 5: maybe I am a bit simple-minded, but I do not understand the difference between “the Fj values and the proportion of introgressed japonica genetic components (alleles)”. Could the Authors explain what is the material difference between these two indices and why they should, or could, diverge? This would greatly help in understanding why a good correlation between them (Figure 3) suggests “that the Fj values … could be used to estimate … the relationships of crop-to-weed introgression with genetic differentiation and genetic diversity”. This whole sentence appears to be trying to convey an important conceptual point, but I do not understand it. Some more explanation would be really appreciated by the general readership.
Response: Thanks for the questions. Actually, the Fj value and the ratio of introgressed japonica alleles are different. The Fj value is the average frequency of the japonica-specific alleles calculated by the following formula (Xjj indicates the homozygous japonica genotype, Xij indicates the heterozygous indica–japonica genotype) [39].
The ratio of introgressed japonica alleles is the proportion of membership of japonica population, which is calculated with InDel genotyping data matrix by STRUCTURE population Q-matrix [50]. The ratio of introgressed japonica alleles represents the level of introgression. Only when Fj value is proved to be positively correlated with the ratio, the Fj values can be used directly to estimate the level of introgression.
Comment 12. Page 8: change “most JS samples weedy rice” to “most JS weedy rice samples”.
Response: Thanks a lot for the comment. We have changed the use of "most JS samples weedy rice" to "most JS weedy rice samples" in the revised manuscript.
Comment 13. Page 8, “the typical indica rice reference varieties showed somehow other genetic components”: I do not know about China indica cultivars, but most Western indica-type cultivars are, in fact, derived from crosses between indica and japonica rices (a few that were obtained years ago by using bridge genotypes), aimed at transferring the indica-type grain in a genetic background more suitable for temperate regions.
Response: Thanks for the comments. Chinese traditional rice varieties generally adopt systematic selection, with a relatively clear genetic background. Meanwhile, we also emphasized in the manuscript that“the typical japonica rice reference varieties showed distinctly unique genetic component (green) with nearly no admixture, although the typical indica rice reference varieties showed somehow other genetic components”. Indica varieties showing other genetic components only indicate indica varieties have greater genetic diversity, which has no influence on the conclusions in this study.
Comment 14. As for the results described on page 11 and Figure 8, it is not clear whether weedy samples with Fj >0.5 matched the expected, ideal trend described in Figure 7.
Response: Thanks a lot for this comment. Many studies indicate that weedy rice in JS is mainly indica type. Due to the self-pollination feature of cultivated rice and weedy rice, there are relatively few intermediate and japonica type of weedy rice in the naturally-sampled weedy rice populations. With only a few decades of crop-to-weed introgression from japonica rice varieties, it is natural that no naturally sampled populations had Fj >0.5.
Comment 15. Figure 8 also shows that InDel are better to highlight the relationship between the level of genetic diversity and crop-to-weed introgression than SSR: could the Authors elaborate a bit on this?
Response: Thanks a lot for this comment. The differences between genetic diversity revealed by SSR and InDel molecular markers can easily be explained by the reasons that the formation of genetic diversity in weedy rice is not only determined by crop-weed introgression involving indica-japonica alleles, but also by other types of alleles that are not associated with the indica and japonica characteristics. We already discussed the differences in the Discussion section of our manuscript.
Comment 16. Page 14: Isn’t the fact that “introgression … can … cause … genetic differentiation” well-known? It is certainly important to note that it was observed in this particular case too, but a suitable reference could be introduced to remind that this is, indeed, a general expectation. Specifically, that genetic introgression between cultivated and weedy plants caused high diversity of weedy red rice has already been reported, for example, by Shivrain et al. (2010. Diversity of weedy red rice (Oryza sativa L.) in Arkansas, USA in relation to weed management. Crop Protection 29:721-730 and references wherein).
Response: Thanks for this comment. The detail mechanism and process of introgression that influences genetic differentiation are not completely known yet. Our point is to use concrete event to explain how increased crop-to-weed introgression can influence the genetic differentiation of weedy rice populations, particularly under human influences. However, the reference provided by the reviewer mainly involves the study of morphological variability in weedy rice, which is different from our findings about genetic differentiation.
Comment 17. Throughout the manuscript, the Authors sometimes refer to old (indica) weedy rices populations as “natural weedy rice populations”: as weedy rice is typical of human agrosystems, it seems a bit forced to consider them as “natural”.
Response: Thanks a lot for the comment. We have modiffied our definition for “ideally-sampled weedy rice population, ISWP” and “naturally-sampled weedy rice population, NSWP”, and used these terminologies consistently throughout the revised manuscript.
Comment 18. Finally, can the present Authors suggest a relationship between the weedy rice populations they studied in this work and the biotypes described by Wang et al. (2023. Occurrence pattern and morphological polymorphism of Chinese weedy rice. J. Integr. Agric. 22:149-169) for the same regions?
Response: Thanks for the comment. However, after carefully reading Wang’s study, we found that the authors classified Chinese weedy rice into three major groups by morphology: multi-tiller (group1), large-leaf (group 2), and large-seed (group 3) weedy rice. Our study is essentially focused on genetic differentiation and diversity of weedy rice from Jiangsu using molecular fingerprints, which is not relevant to Wang’s study.
Comment 19. All in all, the present manuscript presents an interesting case study, with many important data, but it fails to consider important facts (namely, the origin of weedy rices from other species) and alternative hypotheses (that is, weedy rice populations of a different sub-species can be imported as contaminants of the crop seed). I believe, anyway, that the Authors can dutifully account for the missing pieces of information by extensively, and carefully, revise the manuscript.
Response: We indeed appreciate the reviewer’s constructive comments and suggestions. We have carefully revised the manuscript by concentrating more on the results of human influenced weed evolution in agroecosystems.

Round 2
Reviewer 2 Report
I appreciate that the Authors have revised the manuscript. However, I still have concerns.
The main point I raised was that changing indica-japonica characteristics of weedy rice following the rapid change from planting indica to japonica rice varieties could be due to a few seeds of japonica weedy rice contaminating the seed of the imported japonica rice varieties. To this regard, the Authors remarked that they have no evidence about seeds contamination for Jiangsu rice cultivation, because agriculture in this province is highly developed, and most farmers use high-quality commercial “certified seeds” [see reference 32] in which the question of “seeds contamination” is not relevant. Unfortunately, reference [32] makes exactly my case: “hypotheses for the origin of weedy rice strains there have included de-domestication from US cultivars or introduction of an already established weedy rice strain through contamination of seed stock”, and “weed forms are thought to have originated in Asia and been introduced as weeds through accidental import in contaminated seed stocks”. In fact, in many highly developed agricultural systems, wherein farmers use high-quality commercial certified seed, the question of “seeds contamination” is highly relevant. The reason is that it is hard to have high-quality commercial certified seed totally free of weedy rice seeds, so that there is some degree of tolerance for the presence of a few weedy rice seeds in the commercial certified seed. This is the one of the main reasons weedy rice is still a problem in many developed agricultural systems, and not only where farmers use “self-saved seeds for cultivation”.
As remarked in reference [32] as well as in Ziska et al. (2015. Weedy (red) rice: an emerging constraint to global rice production. Adv. Agron. 129:181–228) to understand the origin of red rice in a given agricultural system is a key condition to understand the genetic differentiation and diversity of weedy rice in that system. So, the Authors’ contention that “the origin of weedy rice in Jiangsu” is “not relevant to the manuscript” is wrong, since their manuscript just deals with genetic differentiation and diversity of weedy rice.
The easiest way to be sure that “seeds contamination” is not relevant in the present case is that there is zero tolerance, by law, for the presence of weedy rice seeds in the commercial certified seed used in the Jiangsu province, or (even better) across China, even (particularly) for imported seed. If the Authors can point out that this is the case (about which I have no information) then my concern on this point would immediately drop. If not, then all my previous comments in this regard still await an answer.
In this respect, please, also note that previous remark that “the gradually increased japonica -specific allelic frequency (Fj) in the JS weedy rice samples” (page 10) could also have been observed if “japonica weedy rices, introduced together with the japonica cultivars, had progressively crossed with the old indica weedy rices” is not a “contradictory question” with my remark that “there is a strong reproductive barrier between indica and japonica subspecies, and hybrids between indica and japonica varieties are usually highly sterile”: these are two independent remarks. I first requested that the Authors clarify that there is a strong reproductive barrier between indica and japonica subspecies, but, in some cases, this does not apply (this is obvious given hypothesis (2) in Ziska et al. 2015, namely, weedy rice can have originated from hybridization between wild rice species and cultivated groups). That is, it must be clear that the barrier was the common rule for the rice crop, but in the case of weedy rices, now, the barrier is not strong because weedy rices are a complex of taxonomically not well defined Oryza species, hybrids and special biotypes that can act as bridge genotypes, escaping the strong reproductive barrier existing between the original pure indica and japonica subspecies. Thereafter, I remarked that, if the present Authors cannot exclude that “weed forms … have … been introduced as weeds through accidental import in contaminated seed stocks” [32], then it cannot be excluded that “the gradually increased japonica-specific allelic frequency (Fj) in the JS weedy rice samples” (page 10) could also have been observed if “japonica weedy rices, introduced together with the japonica cultivars, had progressively crossed with the old indica weedy rices”. I acknowledge that the reproductive barrier does no longer occur in some cases (like this one), but I am still concerned about seed contamination.
I am sorry my question that “The simulation performed with ideal weedy rice populations in Figure 5 does not negate that a phenomenon different from crop allelic introgression could cause “genetic differentiation of its conspecific weed” as well” was not clear. I meant to say that, though I agree that “Figure 5 clearly indicates the correlation of allelic introgression with genetic differentiation”, this correlation could be due to either: (i)- “the gradually increased japonica-specific allelic frequency (Fj) in the JS weedy rice samples” (as suggested by the Authors on page 10); as well as to (ii)- my hypothesis that “japonica weedy rices, introduced together with the japonica cultivars, had progressively crossed with the old indica weedy rices”.
In this regard, I am not convinced that “The presence of indica-japonica admixture types in the JS weedy rice samples can also exclude the possible contamination of japonica weedy rice seeds that should be characterized by the pure japonica genotype in the certified commercial cultivar seeds” (lines 494-497). In fact, as I previously remarked, after a few decades of crop and weed coexistence, the “the genetic differentiation of JS weedy rice samples that scattered between the typical reference indica and japonica rice varieties” shown in Fig. 5 could also have been observed if japonica weedy rices, introduced together with the japonica cultivars, had progressively crossed with the old indica weedy rices. The absence of pure japonica genotypes could be due to the fact that the few seeds of japonica weedy rice contaminating the seed of the originally imported japonica rice varieties (coming outside of the Jiangsu province) had produced a few plants that crossed with the old indica weedy rices (already well-adapted to the Jiangsu province) and had subsequently completely substituted with the swarm of indica-japonica progenies that were selected for both their similarity with the now cultivated japonica rice varieties and also for the alleles coming from the old indica weedy rices that improved their adaptability to local conditions. This would be possible if, after the original (contaminated) seed of the (then novel) japonica cultivars had been imported, the seed used in the Jiangsu province was thereafter produced in that same province, thereby selecting against the few original pure japonica weedy rice and in favour of the new swarm of better-adapted indica-japonica progenies. That is, as I previously remarked, the direction of the change of weedy rice in Figure 5 could be from the right to the left instead of from left to right (at least for the swam at the positive loads of the first principal coordinate). Please, note that this is exactly what must have occurred to origin the USA weedy rices if hypothesis (2) in Ziska et al. (2015; namely, weedy rices have originated from hybridization between wild rice species, which are not naturally present in the USA, and cultivated groups) is true. This is why the origin of red rice in a given agricultural system is a key condition to understand the genetic differentiation and diversity of weedy rice in that system.
Moreover, as previously remarked, this whole study, and particularly section 2.4.1 (“Indica-japonica characterization of weedy rice and cultivated rice”), assumes that the weedy rices were Oryza sativa, either indica, japonica, or intermediate. Although weedy rice is often classified as Oryza sativa, other Oryza species such as Oryza rufipogon, Oryza barthii, Oryza nivara, and Oryza longistaminata have also been proposed as sources of weedy rice (Ziska et al. 2015); since, as mentioned above, hypothesis 2 (namely, hybridization between wild rice species and the cultivated crop) appears probable in several cases. Specifically, in regions of tropical Asia where rice is grown in proximity to its wild progenitor (Oryza rufipogon), gene flow from wild populations also contributes to the genetic composition of weed populations ([27,28]; Pusadee et al. 2013. Population structure of the primary gene pool of Oryza sativa in Thailand. Genet. Resour. Crop Evol. 60:335–353; Wang et al. 2017. Asian wild rice is a hybrid swarm with extensive gene flow and feralization from domesticated rice. Genome Res. 27:1029-1038). In general, weedy rice is a complex of taxonomically not well defined Oryza species, hybrids and special biotypes (Kraehmer et al. 2016. Global distribution of rice weeds–a review. Crop Protection 80:73-86). So, how did the present Authors account for the eventuality that portions of the weedy rice populations genome can origin from species different from Oryza sativa?
I still think that the present manuscript presents an interesting case study, with many important data, but it fails to consider important facts (namely, the origin of weedy rices from other species) and alternative hypotheses (that is, weedy rice populations of a different sub-species can be imported as contaminants of the crop seed). I believe, anyway, that the Authors can dutifully account for the missing pieces of information by extensively, and carefully, revise the manuscript.
Author Response
POINT-BY-POINT RESPONSES TO THE COMMENTS OF REVIEWER#2
Comment 1. The main point I raised was that changing indica-japonica characteristics of weedy rice following the rapid change from planting indica to japonica rice varieties could be due to a few seeds of japonica weedy rice contaminating the seed of the imported japonica rice varieties. To this regard, the Authors remarked that they have no evidence about seeds contamination for Jiangsu rice cultivation, because agriculture in this province is highly developed, and most farmers use high-quality commercial “certified seeds” [see reference 32] in which the question of “seeds contamination” is not relevant. Unfortunately, reference [32] makes exactly my case: “hypotheses for the origin of weedy rice strains there have included de-domestication from US cultivars or introduction of an already established weedy rice strain through contamination of seed stock”, and “weed forms are thought to have originated in Asia and been introduced as weeds through accidental import in contaminated seed stocks”. In fact, in many highly developed agricultural systems, wherein farmers use high-quality commercial certified seed, the question of “seeds contamination” is highly relevant. The reason is that it is hard to have high-quality commercial certified seed totally free of weedy rice seeds, so that there is some degree of tolerance for the presence of a few weedy rice seeds in the commercial certified seed. This is the one of the main reasons weedy rice is still a problem in many developed agricultural systems, and not only where farmers use “self-saved seeds for cultivation”.
Response: We understand the reviewer’s concerns about the “seed contamination” issues. However, as we mentioned earlier in our responses to the reviewer, if “a few seeds of japonica weedy rice contaminating the seed of the imported japonica rice varieties” (the possibility cannot be completely ruled out), weedy rice samples with the pure japonica genotype would be identified in the STRUCTURE analysis. But our results based on the STRUCTURE analysis do not show the pure japonica genotype. The reviewer argues that (see the later part in the comment) “The absence of pure japonica genotypes could be due to the fact that the few seeds of japonica weedy rice contaminating the seed of the originally imported japonica rice varieties (coming outside of the Jiangsu province) had produced a few plants that crossed with the old indica weedy rices ...” Again, our explanation is that, even if there are a few japonica weedy rice seeds introduced to Jiangsu rice fields along with the cultivars, the most possible crosses of the few weed rice plants would be easily occurring with the densely cultivated japonica rice varieties closely surrounding these “contaminated” weed rice plants, rather than the far-away, sporadically and scatteredly occurred old indica weedy rice. Therefore, these “contaminated” japonica weedy rice samples, if any, and their hybrids would show the pure japonica genotype. But, our results on the STRUCTURE analysis did not show the pure japonica genotype of weedy rice samples, and do not support that hypothesis.
In addition, here we provided some links of news reports (in Chinese) from the local government in Jiangsu Province, where the government organized serious training on the quality of certified rice seeds and sellers were seriously punished because the certified rice seeds that they sold are found to be low quality or unclean. In Jiangsu, farmers do not save seeds, therefore the purity and credibility of certified seeds can be guaranteed.
https://baijiahao.baidu.com/s?id=1745116821206818066&wfr=spider&for=pc
http://news.jstv.com/a/20230315/1678921823854.shtml
http://coa.jiangsu.gov.cn/art/2020/9/30/art_13248_9520155.html
http://nynct.jiangsu.gov.cn/art/2020/1/10/art_12502_8906910.html
http://nynct.jiangsu.gov.cn/art/2023/3/15/art_11977_10831357.html
https://www.chinaseed114.com/news/9/news_41298.html
Comment 2. As remarked in reference [32] as well as in Ziska et al. (2015. Weedy (red) rice: an emerging constraint to global rice production. Adv. Agron. 129:181–228) to understand the origin of red rice in a given agricultural system is a key condition to understand the genetic differentiation and diversity of weedy rice in that system. So, the Authors’ contention that “the origin of weedy rice in Jiangsu” is “not relevant to the manuscript” is wrong, since their manuscript just deals with genetic differentiation and diversity of weedy rice.
Response: We agree to the reviewer’s point that studying the origin of weedy rice is important. However, this study focused only on the influences of human activities on the rapid evolution of weed rice associated with the extensive change of different rice varieties that promote crop-weed introgression. Our results indicated the change of genetic differentiation and diversity pattern of weedy rice with human influences most likely through crop-to-weed introgression in Jiangsu Province. In such consideration, we feel that no matter which ways weedy rice has originated, the fact of observed human-influenced weedy rice evolution still exists. Therefore, we responded in our previous responses that the study of origin of weedy rice is not directly associated with this study, even though we have other studies addressing the question of weedy rice origins.
Comment 3. The easiest way to be sure that “seeds contamination” is not relevant in the present case is that there is zero tolerance, by law, for the presence of weedy rice seeds in the commercial certified seed used in the Jiangsu province, or (even better) across China, even (particularly) for imported seed. If the Authors can point out that this is the case (about which I have no information) then my concern on this point would immediately drop. If not, then all my previous comments in this regard still await an answer.
Response: About the point of seeds contamination, please see our response to the Comment-1. Again, as we mentioned previously that we provided some news reports (in Chinese) from the local government in Jiangsu Province, where the sellers were seriously punished because the certified rice seeds that they sold are found to be low quality or unclean. In Jiangsu, farmers do not save seeds, therefore the purity and credibility of certified seeds can be guaranteed.
Comment 4. In this respect, please, also note that previous remark that “the gradually increased japonica -specific allelic frequency (Fj) in the JS weedy rice samples” (page 10) could also have been observed if “japonica weedy rices, introduced together with the japonica cultivars, had progressively crossed with the old indica weedy rices” is not a “contradictory question” with my remark that “there is a strong reproductive barrier between indica and japonica subspecies, and hybrids between indica and japonica varieties are usually highly sterile”: these are two independent remarks. I first requested that the Authors clarify that there is a strong reproductive barrier between indica and japonica subspecies, but, in some cases, this does not apply (this is obvious given hypothesis (2) in Ziska et al. 2015, namely, weedy rice can have originated from hybridization between wild rice species and cultivated groups). That is, it must be clear that the barrier was the common rule for the rice crop, but in the case of weedy rices, now, the barrier is not strong because weedy rices are a complex of taxonomically not well defined Oryza species, hybrids and special biotypes that can act as bridge genotypes, escaping the strong reproductive barrier existing between the original pure indica and japonica subspecies. Thereafter, I remarked that, if the present Authors cannot exclude that “weed forms … have … been introduced as weeds through accidental import in contaminated seed stocks” [32], then it cannot be excluded that “the gradually increased japonica-specific allelic frequency (Fj) in the JS weedy rice samples” (page 10) could also have been observed if “japonica weedy rices, introduced together with the japonica cultivars, had progressively crossed with the old indica weedy rices”. I acknowledge that the reproductive barrier does no longer occur in some cases (like this one), but I am still concerned about seed contamination.
Response: Thanks to the reviewer to agree with point that the indica-japonica reproductive barriers are greatly variable. For the point of “seed contamination”, please see our response to the Comment-1.
Comment 5. I am sorry my question that “The simulation performed with ideal weedy rice populations in Figure 5 does not negate that a phenomenon different from crop allelic introgression could cause “genetic differentiation of its conspecific weed” as well” was not clear. I meant to say that, though I agree that “Figure 5 clearly indicates the correlation of allelic introgression with genetic differentiation”, this correlation could be due to either: (i)- “the gradually increased japonica-specific allelic frequency (Fj) in the JS weedy rice samples” (as suggested by the Authors on page 10); as well as to (ii)- my hypothesis that “japonica weedy rices, introduced together with the japonica cultivars, had progressively crossed with the old indica weedy rices”.
Response: The reviewer insists to hypothesize that “japonica weedy rices, introduced together with the japonica cultivars, had progressively crossed with the old indica weedy rices.” Firstly, the reviewer agrees that crosses (introgression) between japonica and indica rice/weedy rice samples are highly possible. Secondly, as we already explained in our response to the comment-1 that even if there are a few japonica weedy rice seeds introduced to Jiangsu rice fields along with the cultivars, the most possible crosses of the few weed rice plants would be easily occurring with the densely cultivated japonica rice varieties closely surrounding these “contaminated” weed rice plants, rather than the far-away, sporadically and scatteredly occurred indica weedy rice. Therefore, these “contaminated” japonica weedy rice samples, if any, and their hybrids would show the pure japonica genotype. But, our results on the STRUCTURE analysis did not show the pure japonica genotype of weedy rice samples, and do not support that hypothesis. We should not negate the experimental results by only hypothesis. In addition, for the point of “seed contamination”, please see our response to the Comment-1.
Comment 6. In this regard, I am not convinced that “The presence of indica-japonica admixture types in the JS weedy rice samples can also exclude the possible contamination of japonica weedy rice seeds that should be characterized by the pure japonica genotype in the certified commercial cultivar seeds” (lines 494-497). In fact, as I previously remarked, after a few decades of crop and weed coexistence, the “the genetic differentiation of JS weedy rice samples that scattered between the typical reference indica and japonica rice varieties” shown in Fig. 5 could also have been observed if japonica weedy rices, introduced together with the japonica cultivars, had progressively crossed with the old indica weedy rices. The absence of pure japonica genotypes could be due to the fact that the few seeds of japonica weedy rice contaminating the seed of the originally imported japonica rice varieties (coming outside of the Jiangsu province) had produced a few plants that crossed with the old indica weedy rices (already well-adapted to the Jiangsu province) and had subsequently completely substituted with the swarm of indica-japonica progenies that were selected for both their similarity with the now cultivated japonica rice varieties and also for the alleles coming from the old indica weedy rices that improved their adaptability to local conditions. This would be possible if, after the original (contaminated) seed of the (then novel) japonica cultivars had been imported, the seed used in the Jiangsu province was thereafter produced in that same province, thereby selecting against the few original pure japonica weedy rice and in favour of the new swarm of better-adapted indica-japonica progenies. That is, as I previously remarked, the direction of the change of weedy rice in Figure 5 could be from the right to the left instead of from left to right (at least for the swam at the positive loads of the first principal coordinate). Please, note that this is exactly what must have occurred to origin the USA weedy rices if hypothesis (2) in Ziska et al. (2015; namely, weedy rices have originated from hybridization between wild rice species, which are not naturally present in the USA, and cultivated groups) is true. This is why the origin of red rice in a given agricultural system is a key condition to understand the genetic differentiation and diversity of weedy rice in that system.
Response: For this point, please see the explanations in our response to comment-5. However, we modified the relevant sentences in lines 494-497 to make our statement clearer.
Comment 7. Moreover, as previously remarked, this whole study, and particularly section 2.4.1 (“Indica-japonica characterization of weedy rice and cultivated rice”), assumes that the weedy rices were Oryza sativa, either indica, japonica, or intermediate. Although weedy rice is often classified as Oryza sativa, other Oryza species such as Oryza rufipogon, Oryza barthii, Oryza nivara, and Oryza longistaminata have also been proposed as sources of weedy rice (Ziska et al. 2015); since, as mentioned above, hypothesis 2 (namely, hybridization between wild rice species and the cultivated crop) appears probable in several cases. Specifically, in regions of tropical Asia where rice is grown in proximity to its wild progenitor (Oryza rufipogon), gene flow from wild populations also contributes to the genetic composition of weed populations ([27,28]; Pusadee et al. 2013. Population structure of the primary gene pool of Oryza sativa in Thailand. Genet. Resour. Crop Evol. 60:335–353; Wang et al. 2017. Asian wild rice is a hybrid swarm with extensive gene flow and feralization from domesticated rice. Genome Res. 27:1029-1038). In general, weedy rice is a complex of taxonomically not well defined Oryza species, hybrids and special biotypes (Kraehmer et al. 2016. Global distribution of rice weeds–a review. Crop Protection 80:73-86). So, how did the present Authors account for the eventuality that portions of the weedy rice populations genome can origin from species different from Oryza sativa?
Response: The taxonomic classification of weedy rice has been recorded in many articles, which is not the research problem in this study. The fact is that weedy rice found in Jiangsu Province (no natural distribution of any wild rice) shares a very close genetic relationship with cultivated rice, and the detected crop-to-weed introgression can influence the evolution of weedy rice, regardless of its taxonomic status. Although weedy rice is a complex in terms of its taxonomy, the objective of this study is to provide a simple case to emphasize that human activities and disturbances can largely influence the evolution of weeds in agroecosystems.
Comment 8. I still think that the present manuscript presents an interesting case study, with many important data, but it fails to consider important facts (namely, the origin of weedy rices from other species) and alternative hypotheses (that is, weedy rice populations of a different sub-species can be imported as contaminants of the crop seed). I believe, anyway, that the Authors can dutifully account for the missing pieces of information by extensively, and carefully, revise the manuscript.
Response: Thanks for the positive part of the comment. For the point of the seed contamination, please see our responses to comment-1 and comment-5. For the point of weedy rice origin, please see our responses to comment-2 and to comment-7.

Round 3
Reviewer 2 Report
I agree with the Authors that «if there are a few japonica weedy rice seeds introduced to Jiangsu rice fields along with the cultivars, the most possible crosses of the few weed rice plants would be easily occurring with the densely cultivated japonica rice varieties closely surrounding these “contaminated” weed rice plants, rather than the far-away, sporadically and scatteredly occurred old indica weedy rice», yet, the cross with old indica weedy rice was possible. It would also be easy because, often, weedy rices are early flowering with respect to the rice crop, so that, their inter-cross would be facilitated.
It is true that «these “contaminated” japonica weedy rice samples, if any, and their hybrids would show the pure japonica genotype», unless they had disappeared at the time of the analysis because, as I have already said, the old indica weedy rices were better suited to the local environment, which, therefore, acted by “selecting against the few original pure japonica weedy rice and in favour of the new swarm of better-adapted indica-japonica progenies”. This is entirely consistent with the literature I have previously mentioned. It would be, thus, obvious that “the STRUCTURE analysis did not show the pure japonica genotype of weedy rice samples”, since they had already disappeared by the time this study was done.
The study of origin of weedy rice is directly associated with this study because we are discussing whether weedy rice in the Jiangsu province originated from a cross either between old indica weedy rices and new japonica crop cultivars, or between old indica weedy rices and new japonica weedy rices. These are two different origins.
I hope it is clear that I am not saying that the Authors’ results are wrong. What I am insisting on is that the hypothesis that “japonica weedy rices, introduced together with the japonica cultivars, had progressively crossed with the old indica weedy rices” should be carefully considered, not quickly dismissed. As I have already stated, the easiest «way to be sure that “seeds contamination” is not relevant in the present case is that there is zero tolerance, by law, for the presence of weedy rice seeds in the commercial certified seed used in the Jiangsu province, or (even better) across China, even (particularly) for imported seed». As the Authors reports that “sellers were seriously punished because the certified rice seeds that they sold are found to be low quality or unclean”, this means, I suppose, that such law exists. So, please, clearly state, in the manuscript, that the Authors deem that seeds of japonica weedy rices could not be introduced together with the japonica cultivars because there is zero tolerance, by law, for the presence of weedy rice seeds in the commercial certified seed used in the Jiangsu province. In this way, the present contention will be immediately solved.
Besides, as the Authors performed a specific “Indica-japonica characterization of weedy rice and cultivated rice”, the fact that “weedy rice is a complex of taxonomically not well defined Oryza species, hybrids” (Kraehmer et al. 2016) should not be dismissed. This is why I asked how the Authors did “account for the eventuality that portions of the weedy rice populations genome can origin from species different from Oryza sativa”. If the Authors know “that weedy rice found in Jiangsu Province … shares a very close genetic relationship with cultivated rice”, that is the answer; but it should be made clear in the manuscript. That is, if the Authors made a comparison of the genome of the Jiangsu weedy rice with the genomes of several Oryza species and they found that it matched with Oryza sativa only (maybe, 99%), this would be a clear demonstration that genome portions from species different from Oryza sativa are not relevant to this study. This is not a tough requirement on my side, I think. In other words, the fact that “weedy rice is genetically closely associated with its cultivated counterparts cooccurring in the same regions [21,23,27,31]” (lines 484-485) must be a premise to this study, not a conclusion of it. Accordingly, the fact that “the Fj values of the typical japonica rice varieties and weedy rice samples from NEC were equal to 1 or close to 1” (lines 293-294), would be a confirmation of previous results.
Author Response
POINT-BY-POINT RESPONSES TO THE COMMENTS OF REVIEWER#2
Comment 1. I agree with the Authors that «if there are a few japonica weedy rice seeds introduced to Jiangsu rice fields along with the cultivars, the most possible crosses of the few weed rice plants would be easily occurring with the densely cultivated japonica rice varieties closely surrounding these “contaminated” weed rice plants, rather than the far-away, sporadically and scatteredly occurred old indica weedy rice», yet, the cross with old indica weedy rice was possible. It would also be easy because, often, weedy rices are early flowering with respect to the rice crop, so that, their inter-cross would be facilitated.
Response: Thanks for reviewer’s comment. We agree to the reviewer’s point about the generally earlier flowering time of weedy rice plants than their cooccurring cultivated rice. Actually, inter-crosses between weedy rice and its cooccurring rice cultivars have been frequently reported both in the gene flow experiments [e.g., the reference 16, 62-64 in our revised manuscript, and Chen et al. 2004; Goulart et al. 2015; Nam et al., 2019] and under natural conditions [e.g., the references 17, 20, 21 in this manuscript, and Shivrain et al. 2007, 2008; Vigueira et al., 2018]. Somehow, we have not yet found a published article reporting or emphasizing the crosses between the two types of weedy rice (including indica and japonica weedy rice). Similarly, we could not provide solid data or evidence to prove introgression between the indica and japonica types of weedy rice. Therefore, we hesitated to make the statement of weed-to-weed introgression in this manuscript.
Hypothetically, it is possible to have crosses between newly introduced (contaminated) japonica weedy rice and old indica weedy rice. However, given the low density of weedy rice (ca. 0.5-1.0 per 100 m2, we have added the information in the modified manuscript in M&M) in the Jiangsu rice fields, in addition to the low outcrossing rate of weedy rice [<1%; please see the references 62-64 in our revised manuscript; also, from Chen et al. 2004; Shivrain et al. 2007, 2008; Goulart et al. 2015; Nam et al. 2019], the opportunity of crosses between newly introduced (contaminated) japonica weedy rice and old indica weedy rice would be very low. On the other hand, the possibility of crosses between weedy rice and its densely surrounded cultivated rice should be much greater than that between two types of weedy rice. Possibly, we have not sampled the contaminated japonica weedy rice or these hybrids in our sample collection.
Even though there are a few possible contaminated japonica weedy rice seed plants crossed with old indica weedy rice, the frequency of the pure japonica weedy rice and its hybrid progeny would be extremely low. Most likely, the extremely low mixed seeds, if any, would not change the general conclusion of this study about the human influenced weed evolution. However, following the reviewer’s suggestion, we have added some information about the possibility of mixed weedy rice seeds in the revised manuscript.
Supporting references:
- Chen et al. 2004. Gene flow from cultivated rice (Oryza sativa) to its weedy and wild relatives. Annals of Botany, 93: 67-73.
- Shivrain, et al. 2007. Gene flow between ClearfieldTM rice and red rice. Crop Protection, 26: 349-356.
- Shivrain, et al. 2008. Maximum outcrossing rate and genetic compatibility between red rice (Oryza sativa) biotypes and ClearfieldTM Weed Science, 56: 807-813.
- Goulart, et al. 2015. Detecting gene flow from ALS-resistant hybrid and inbred rice to weedy rice using single plant pollen donors. Experimental Agriculture, 52: 237-250.
- Vigueira, et al. 2018. Pink-awned weedy rice (Oryza sativa): a potential conduit for gene exchange in rice agro-ecosystems. Weed Research, 58: 369-378.
- Nam, et al. 2019. Gene flow from transgenic PPO-inhibiting herbicide-resistant rice to weedy rice, and agronomic performance by their hybrids. Journal of Plant Biology, 62: 286-296
Comment 2. It is true that «these “contaminated” japonica weedy rice samples, if any, and their hybrids would show the pure japonica genotype», unless they had disappeared at the time of the analysis because, as I have already said, the old indica weedy rices were better suited to the local environment, which, therefore, acted by “selecting against the few original pure japonica weedy rice and in favour of the new swarm of better-adapted indica-japonica progenies”. This is entirely consistent with the literature I have previously mentioned. It would be, thus, obvious that “the STRUCTURE analysis did not show the pure japonica genotype of weedy rice samples”, since they had already disappeared by the time this study was done.
Response: Thanks for the reviewer’s concern, our explanation to this comment is that if weedy rice seed contamination exists systematically in Jiangsu rice planting regions where we collected our research materials, such contamination of weedy rice seeds should happen successively every year, but should not happen only in a particular year. Therefore, the continued introduction of weedy rice seeds would not have disappeared and should be collected in our samples. However, as indicated, our results from the STRUCTURE analysis did not show contaminated pure japonica weedy rice seeds in >1000 samples used in this study.
Comment 3. The study of origin of weedy rice is directly associated with this study because we are discussing whether weedy rice in the Jiangsu province originated from a cross either between old indica weedy rices and new japonica crop cultivars, or between old indica weedy rices and new japonica weedy rices. These are two different origins.
Response: Thanks a lot for reminding us to discuss the possibility of crosses between old indica weedy rice and new japonica weedy rice in addition to old indica weedy rice and new japonica cultivars. Our conclusion is that weedy rice with indica-japonica characteristics in our samples should have originated from crosses between old indica weedy rice and new japonica cultivars. The reason for such a conclusion is based on the following three facts:
- The density of weedy rice in Jiangsu rice fields is low (ca. 0.5-1.0 per 100 m2) at the time when weedy rice was sampled, compared with cultivated rice that is densely cultivated in the rice fields. Therefore, the possibility of crosses between scattered weedy rice plants should be much lower than those between weedy and cultivated rice plants.
- Given that the density of cultivated rice is much higher than that of weedy rice in the same rice fields. Therefore, the possibility of crosses between weedy and cultivated rice should be much greater than that those between weedy rice plants.
- As a self-pollination and wind-pollination taxon, weedy rice has a very low outcrossing rate [<1%; reference 62-64 in our revised manuscript; and previous references Shivrain, 2007; Goulart, 2015; Nam, 2019] and the distance of pollen mediated gene flow is generally short in rice (<10m) [Messeguer, 2003; Rong et al., 2006; Goulart et al., 2015]. Therefore, the inter-crosses between relatively distantly distributed japonica and indica weedy rice would be extremely low, compared with the crosses between close weedy and cultivated rice. From statistical point of view, the possibility of crosses between weedy rice and cultivated rice is much greater than that of the crosses between weedy rice plants.
Supporting references:
Messeguer. 2003. Gene flow assessment in transgenic plants. Plant Cell, Tissue and Organ Culture, 73:201-212. Doi: 10.1023/A:1023007606621
Rong, et al. 2007. Dramatic reduction of crop-to-crop gene flow within a short distance from transgenic rice fields. New Phytologist, 173:346-353. Doi: 10.1111/j.1469-8137.2006.01906.x
Goulart, et al. 2015. Detecting gene flow from ALS-resistant hybrid and inbred rice to weedy rice using single plant pollen donors. Experimental Agriculture, 52:237-250. Doi: 10.1017/S0014479715000058
Comment 4. I hope it is clear that I am not saying that the Authors’ results are wrong. What I am insisting on is that the hypothesis that “japonica weedy rices, introduced together with the japonica cultivars, had progressively crossed with the old indica weedy rices” should be carefully considered, not quickly dismissed. As I have already stated, the easiest «way to be sure that “seeds contamination” is not relevant in the present case is that there is zero tolerance, by law, for the presence of weedy rice seeds in the commercial certified seed used in the Jiangsu province, or (even better) across China, even (particularly) for imported seed». As the Authors reports that “sellers were seriously punished because the certified rice seeds that they sold are found to be low quality or unclean”, this means, I suppose, that such law exists. So, please, clearly state, in the manuscript, that the Authors deem that seeds of japonica weedy rices could not be introduced together with the japonica cultivars because there is zero tolerance, by law, for the presence of weedy rice seeds in the commercial certified seed used in the Jiangsu province. In this way, the present contention will be immediately solved.
Response:
We appreciate the reviewer for his positive comments about our results. We have to confess that we have no right to state the zero tolerance by law for weed contamination. We agree to the reviewer’s point that it is impossible to completely reject japonica weedy rice contamination, and that contaminated japonica weedy rice, if any, can cross with the old indica weedy rice. However, results based on the STRUCTURE analysis indicated extremely low possibility of contaminated seeds in our studied samples. About this point of seed contamination, we have modified some sentences in the Discussion section in the revised manuscript by adding the statement that “there is a possibility of weedy rice seed contamination with the introduced japonica rice varieties in practical rice production.”
In addition, about the possibility of crosses between different types of weedy rice, we have checked literature databases but we have not found any published paper specifically reporting crosses between the two types of weedy rice (including indica and japonica weedy rice). Therefore, we hesitated to state the origin of indica-japonica types of weedy rice is from crosses between old indica weedy rice and new japonica weedy rice. Anyhow, as we indicated earlier that even though there are a few contaminated japonica weedy rice seed plants crossed with old indica weedy rice, that may not change the general conclusion of this study about the human influenced weed evolution. However, we added some sentences to address this in the revised manuscript.
Comment 5. Besides, as the Authors performed a specific “Indica-japonica characterization of weedy rice and cultivated rice”, the fact that “weedy rice is a complex of taxonomically not well defined Oryza species, hybrids” (Kraehmer et al. 2016) should not be dismissed. This is why I asked how the Authors did “account for the eventuality that portions of the weedy rice populations genome can origin from species different from Oryza sativa”. If the Authors know “that weedy rice found in Jiangsu Province … shares a very close genetic relationship with cultivated rice”, that is the answer; but it should be made clear in the manuscript. That is, if the Authors made a comparison of the genome of the Jiangsu weedy rice with the genomes of several Oryza species and they found that it matched with Oryza sativa only (maybe, 99%), this would be a clear demonstration that genome portions from species different from Oryza sativa are not relevant to this study. This is not a tough requirement on my side, I think. In other words, the fact that “weedy rice is genetically closely associated with its cultivated counterparts cooccurring in the same regions [21,23,27,31]” (lines 484-485) must be a premise to this study, not a conclusion of it. Accordingly, the fact that “the Fj values of the typical japonica rice varieties and weedy rice samples from NEC were equal to 1 or close to 1” (lines 293-294), would be a confirmation of previous results.
Response:
About the point of genetic relationships between weedy rice and cultivated rice, there are already many research articles reporting the dedomestication origins of weedy rice in Jiangsu province, as a result weedy rice there is genetically similar to the cultivated rice, rather than other Oryza species (Oryza rufipogon, Oryza nivara, etc.) [please see the references 34,42 in our vised manuscript; and Li et al., 2017; Qiu et al., 2017, 2020; Zhu et al., 2021].
About the point including the NEC and GD weedy rice samples that are used as references for the japonica type (NEC, Fj close to 1) and indica type (GD, Fj close to 0) of weedy rice. We have modified the relevant sentences in the Material and Methods, Discussion sections.
Supporting references:
Li et al. 2017. Signatures of adaptation in the weedy rice genome. Nature Genetics, 49: 811-814.
Qiu et al. 2017. Genomic variation associated with local adaptation of weedy rice during de-domestication. Nature Communicaitons, 8: 15323.
Qiu et al. 2020. Diverse genetic mechanisms underlie worldwide convergent rice feralization. Genome Biology. 21:70.
Zhu et al. 2021. Key roles of de-domestication and novel mutation in origin and diversification of global weedy rice. Biology, 10: 828.

Round 4
Reviewer 2 Report
Although I'd like to see some more deep discussion about the hypothesis that “japonica weedy rices, introduced together with the japonica cultivars, had progressively crossed with the old indica weedy rices”, my main concern was that this hypothesis should be carefully considered and not quickly dismissed. As already mentioned, reference [32] makes exactly my case: “hypotheses for the origin of weedy rice strains ... included ... introduction of an already established weedy rice strain through contamination of seed stock”, and “weed forms are thought to have originated in Asia and been introduced as weeds through accidental import in contaminated seed stocks”. The Authors have now given enough consideration to such hypothesis (and, most important, they give practical reasons to believe this hypothesis has a low probability of occurrence). Therefore, I do not insist more on it. Although, I recommend the Authors to stress that "results from the STRUCTURE analysis did not show ... ANY pure japonica weedy rice seeds in >1000 samples used in this study".
I also recommend, again, that the Authors make clear, in the Introduction, that since in these "regions (e.g., northeastern China) ... no wild Oryza species are distributed [21,29,31]" (lines 110-111) "as a result weedy rice there is genetically similar to the cultivated rice, rather than other Oryza species (Oryza rufipogon, Oryza nivara, etc.) [please see the references 34,42 in our vised manuscript; and Li et al., 2017; Qiu et al., 2017, 2020; Zhu et al., 2021]".
Line 480: 'owing'.
As previously remarked, I think that the present manuscript presents an interesting case study.
Author Response
POINT-BY-POINT RESPONSES TO THE COMMENTS OF REVIEWER#2
Comment 1. Although I'd like to see some more deep discussion about the hypothesis that “japonica weedy rices, introduced together with the japonica cultivars, had progressively crossed with the old indica weedy rices”, my main concern was that this hypothesis should be carefully considered and not quickly dismissed. As already mentioned, reference [32] makes exactly my case: “hypotheses for the origin of weedy rice strains ... included ... introduction of an already established weedy rice strain through contamination of seed stock”, and “weed forms are thought to have originated in Asia and been introduced as weeds through accidental import in contaminated seed stocks”. The Authors have now given enough consideration to such hypothesis (and, most important, they give practical reasons to believe this hypothesis has a low probability of occurrence). Therefore, I do not insist more on it. Although, I recommend the Authors to stress that "results from the STRUCTURE analysis did not show ... ANY pure japonica weedy rice seeds in >1000 samples used in this study".
Response: We indeed appreciate the reviewer’s positive comments to our revised manuscript.
Comment 2. I also recommend, again, that the Authors make clear, in the Introduction, that since in these "regions (e.g., northeastern China) ... no wild Oryza species are distributed [21,29,31]" (lines 110-111) "as a result weedy rice there is genetically similar to the cultivated rice, rather than other Oryza species (Oryza rufipogon, Oryza nivara, etc.) [please see the references 34,42 in our vised manuscript; and Li et al., 2017; Qiu et al., 2017, 2020; Zhu et al., 2021]".
Response: Thanks for the this recommendation and we have added relevant sentences in the Introduction of the revised manuscript.
Comment 3. Line 480: 'owing'.
Response: Thanks for pointing out this error and we have corrected it in the revised manuscript.
Comment 4. As previously remarked, I think that the present manuscript presents an interesting case study.
Response: Thanks a lot for the reviewer’s positive comments. Also, we whole-heartly appreciate the reviewer’s great efforts to improve the manuscript.
